# EvoGM: Learning to Merge LLMs via Evolutionary Generative Optimization

**Tao Jiang** [1 2] **Xinmeng Yu** [1 2] **Chenhao Yi** [3 2] **Yiling Wu** [2] **Yan Li** [2] **Ran Cheng** [4 5 6] **Dongmei Jiang** [2 †] **Jianguo Zhang** [1 2 7 †]

## Abstract

Evolutionary model merging provides a powerful framework for the automated, training-free composition of LLMs through parameter-space search. However, existing methods predominantly rely on stochastic, hand-crafted operators that overlook the underlying performance landscape of the coefficient space. We propose Evolutionary Generative Merging (EvoGM), a framework that transcends manual heuristics by employing learnable generative modeling to optimize merging coefficients. Specifically, EvoGM features a dual-generator architecture with cycle-consistent learning to adaptively sample and refine promising merging candidates. By constructing winner-loser pairs from historical search trajectories, our framework effectively captures high-performance parameter distributions and maximizes data efficiency. This generative process is seamlessly integrated into a multi-round evolutionary pipeline, where elite merged models iteratively serve as new expert foundations. Extensive experiments across diverse benchmarks demonstrate that EvoGM significantly outperforms state-of-the-art baselines, exhibiting robust performance on both seen and unseen tasks. Code and data are available at https://github.com/JiangTao97/evogm.

---

[†]Corresponding authors. [1]Research Institute of Trustworthy Autonomous Systems and Department of Computer Science and Engineering, Southern University of Science and Technology [2]Pengcheng Laboratory [3]University of Chinese Academy of Sciences [4]Department of Data Science and Artificial Intelligence, The Hong Kong Polytechnic University [5]Hong Kong Polytechnic University Shenzhen Research Institute [6]Hong Kong Polytechnic University-Daya Bay Technology and Innovation Research Institute [7]Guangdong Provincial Key Laboratory of Brain-inspired Intelligent Computation. Correspondence to: Dongmei Jiang <jiangdm@pcl.ac.cn>, Jianguo Zhang <zhangjg@sustech.edu.cn>.

*Proceedings of the 43rd International Conference on Machine Learning*, Seoul, South Korea. PMLR 306, 2026. Copyright 2026 by the author(s).

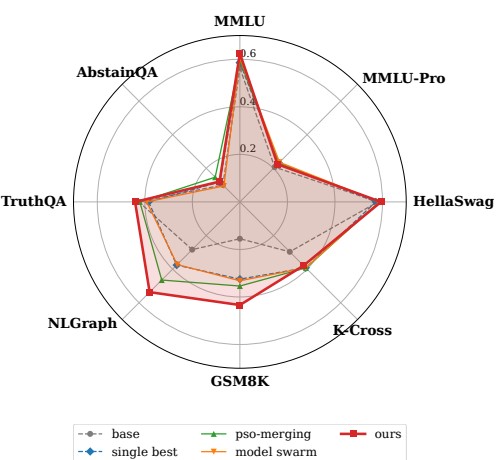

*Figure 1.* Single-task performance comparison of 10 fine-tuned Qwen2.5-1.5B models. The proposed method outperforms baseline models on almost all targeted tasks.

## 1. Introduction

The prevailing paradigm of large language models (LLMs) relies on large-scale pretraining followed by task-specific adaptation, yet the rapid growth in model size makes full-parameter fine-tuning increasingly impractical under realistic computational and data constraints (Hu et al., 2022; Li & Liang, 2021; Lv et al., 2024). In this context, a key challenge is how to accumulate and compose model capabilities without repeatedly training new models. Model merging addresses this challenge by directly integrating multiple expert models into a single one, enabling efficient knowledge transfer and capability composition at no additional training cost, and has therefore emerged as a promising mechanism for scalable and sustainable model improvement (Lu et al., 2024a; Yang et al., 2026; Zheng et al., 2025).

Most existing model merging methods rely on manually designed heuristics, including linear averaging (Ilharco et al., 2023; Jin et al., 2025; Tang et al., 2024; Wortsman et al., 2022; Yu et al., 2024a), spherical interpolation (Jiang et al., 2025), or the use of scaling and sparsifying schemes to mitigate parameter interference (Ma et al., 2025; Sun et al., 2025b; Yang et al., 2025). These approaches are largely task-agnostic and lack the ability to adapt their merging strategies according to validation objectives. To address this limitation,

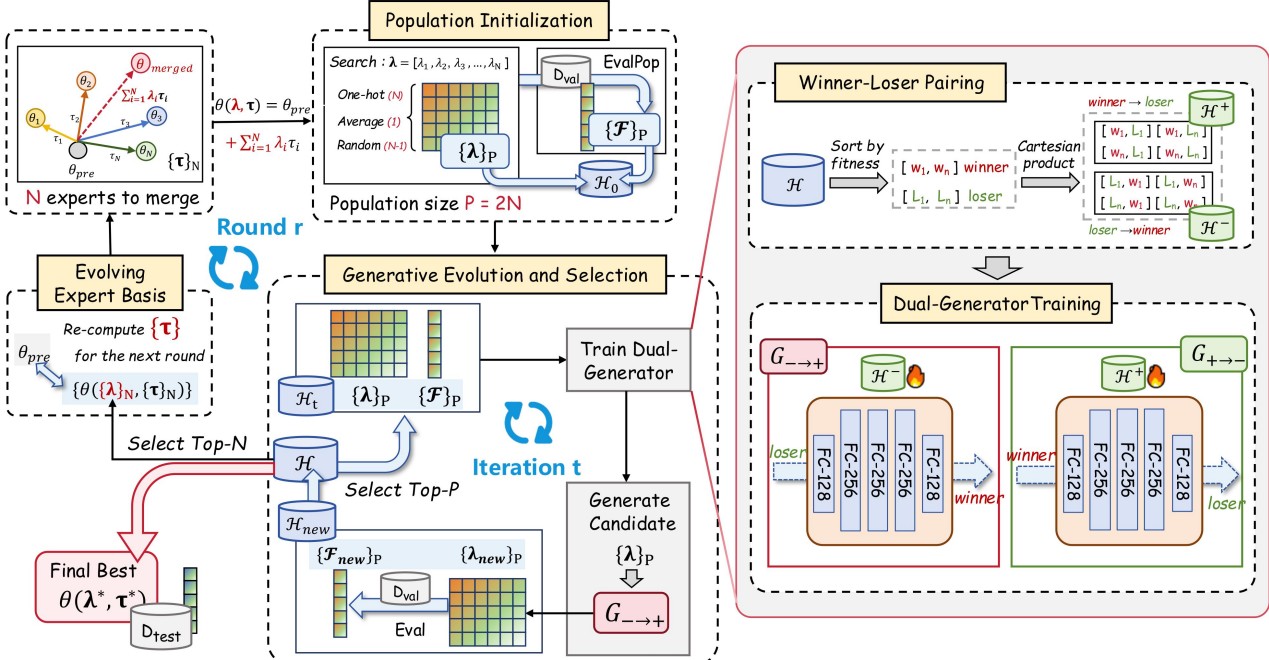

*Figure 2.* **Overview of EvoGM.** We optimize merging coefficients $\boldsymbol{\lambda} \in \mathbb{R}^N$ for task-vector merging. (1) *Population Initialization:* initialize a diverse population of $\boldsymbol{\lambda}$ (average merge, one-hot, random) and evaluate on $\mathcal{D}_{val}$ to build the history set $\mathcal{H} = \{(\boldsymbol{\lambda}, f(\boldsymbol{\lambda}))\}$. (2) *Winner–Loser Pairing:* split $\mathcal{H}$ into winners $\mathcal{H}^+$ and losers $\mathcal{H}^-$. (3) *Dual-Generator Training:* train $(G_{-\to+}, G_{+\to-})$ with cycle-consistency and optimization-guided losses. (4) *Generative Evolution & Selection:* apply $G_{-\to+}$ to generate new candidates and select improved $\boldsymbol{\lambda}$ by evaluation. (5) *Evolving Expert Basis:* periodically refresh the expert pool via elite merges and recompute $\{\boldsymbol{\tau}_i\}_{i=1}^N$, returning the best $\boldsymbol{\lambda}^*$ and merged model $\theta(\boldsymbol{\lambda}^*)$.

more recent studies have explored optimization- or search-based formulations (Li et al., 2025a; Liu et al., 2025; Wang et al., 2020). Furthermore, evolutionary model merging has emerged as a practical alternative, enabling gradient-free population-based search and demonstrating competitive performance across a wide range of tasks (Akiba et al., 2025; Inoshita et al., 2024; Mencattini et al., 2025).

Despite their empirical success, existing evolutionary model merging methods largely operate over an unstructured search space and depend on stochastic perturbations, resulting in low search efficiency under realistic deployment constraints with small validation sets and limited evaluation budgets. In optimization problems characterized by similarly sparse and expensive feedback, generative optimization approaches have been shown to substantially reduce search randomness by learning structured proposal distributions from limited evaluative signals (Gao et al., 2024; Jiang et al., 2026; Wang et al., 2025). Motivated by these advances, we argue that model merging should be formulated as a data-driven and learnable search process rather than a purely stochastic one. By treating validation performance as a reward signal, a generative model can capture the distribution of high-performing merging configurations, enabling

structured exploration of the merging space and improved generalization.

Building on this insight, we propose Evolutionary Generative Merging (EvoGM), a unified framework that integrates generative modeling into evolutionary model merging. EvoGM formulates the search over merging coefficients as adaptive sampling from a learned probability distribution, replacing static, hand-crafted mutation operators. The framework employs a dual-generator architecture with cycle-consistent constraints to capture the geometric structure of high-performance regions while preserving diversity. To effectively leverage past search experience, we introduce a winner-loser preference mechanism that constructs comparative signals from historical trajectories, enabling the generative model to distinguish promising directions from ineffective ones. Embedded within a multi-round evolutionary pipeline, elite merged models iteratively serve as new expert foundations, allowing the search strategy and model capabilities to co-evolve. Extensive experiments demonstrate that EvoGM achieves significantly improved efficiency and robustness, consistently outperforming prior approaches on both seen and unseen tasks.

Our main contributions are summarized as follows:

- We reformulate evolutionary model merging as a learnable generative task, enabling adaptive characterization of high-performance regions.

- We introduce a dual-generator architecture with cycle-consistency and a winner-loser preference strategy to maximize data utilization from sparse feedback, enabling efficient model synthesis for unseen tasks.

- Extensive experiments demonstrate that EvoGM significantly outperforms state-of-the-art baselines across diverse benchmarks and model families.

## 2. Methodology

### 2.1. Problem Formulation

We formulate model merging as an optimization task that seeks the optimal coefficients for combining task vectors. Let $\boldsymbol{\theta}_{pre} \in \mathbb{R}^d$ denote the parameters of a pre-trained base model, and $\{\boldsymbol{\theta}_i\}_{i=1}^N$ be a set of $N$ expert models fine-tuned from $\boldsymbol{\theta}_{pre}$ for specific tasks. Following the principles of task arithmetic, we define the task vector for each expert as $\boldsymbol{\tau}_i = \boldsymbol{\theta}_i - \boldsymbol{\theta}_{pre}$. The merged model is parameterized by a coefficient vector $\boldsymbol{\lambda} = [\lambda_1, \ldots, \lambda_N]^\top \in \mathbb{R}^N$, where the dimensionality of the optimization space corresponds directly to the number of expert models $N$. This results in parameters $\boldsymbol{\theta}(\boldsymbol{\lambda}) = \boldsymbol{\theta}_{pre} + \sum_{i=1}^N \lambda_i \boldsymbol{\tau}_i$. Given a validation set $\mathcal{D}_{val}$ and an evaluation metric $f(\cdot)$, our objective is to find the optimal coefficients $\boldsymbol{\lambda}^* = \arg\max_{\boldsymbol{\lambda}} f(\boldsymbol{\theta}(\boldsymbol{\lambda}); \mathcal{D}_{val})$. In this framework, the optimization is restricted to the low-dimensional vector $\boldsymbol{\lambda} \in \mathbb{R}^N$.

**Limitations** Finding $\boldsymbol{\lambda}^*$ is non-trivial. Grid search is computationally prohibitive due to the exponential growth of the search space with $N$, while existing evolutionary methods rely on stochastic perturbations and manual heuristics. These approaches often fail to account for the underlying performance distribution, frequently leading to sub-optimal results in complex merging scenarios.

> *Can we transcend manual search heuristics by learning to locate optimal merging configurations?*

### 2.2. Evolutionary Generative Merging

To address the challenge, we propose to transform the search process into a learnable generative task. Instead of directly perturbing $\boldsymbol{\lambda}$ through random mutations, EvoGM characterizes the distribution of optimal merging coefficients by modeling the relationship between historical search trajectories and their corresponding performance on $\mathcal{D}_{val}$. The EvoGM framework is illustrated in Figure 2, with the full procedure detailed in Algorithm 1.

---

**Algorithm 1** EvoGM: Evolutionary Generative Merging

**Input:** Base model $\boldsymbol{\theta}_{pre}$, initial experts $\{\boldsymbol{\theta}_i\}_{i=1}^N$, validation set $\mathcal{D}_{val}$, population size $P$, number of rounds $R$, number of iterations per round $T$
**Output:** Optimal merging coefficients $\boldsymbol{\lambda}^*$
Compute initial task vectors: $\boldsymbol{\tau}_i \leftarrow \boldsymbol{\theta}_i - \boldsymbol{\theta}_{pre}$
**for** round $r = 1$ **to** $R$ **do**
  $\mathcal{P} \leftarrow \text{InitPop}(P)$; $\mathcal{F} \leftarrow f(\boldsymbol{\theta}(\mathcal{P}), \mathcal{D}_{val})$
  $\mathcal{H} \leftarrow \{(\boldsymbol{\lambda}^{(p)}, \mathcal{F}^{(p)})\}_{p=1}^P$
  Reset Dual-Generators $G_{-\rightarrow+}, G_{+\rightarrow-}$
  **for** iteration $t = 1$ **to** $T$ **do**
    $\mathcal{H}^+, \mathcal{H}^- \leftarrow \text{PairWinnersLosers}(\mathcal{H})$
    Train Dual-Generators $G_{-\rightarrow+}, G_{+\rightarrow-}$
    $\boldsymbol{\lambda}_{new} \leftarrow G_{-\rightarrow+}(\mathcal{H}^+)$
    $\mathcal{F}_{new} \leftarrow f(\boldsymbol{\theta}(\boldsymbol{\lambda}_{new}), \mathcal{D}_{val})$
    $\mathcal{H} \leftarrow \mathcal{H} \cup \{(\boldsymbol{\lambda}_{new}, \mathcal{F}_{new})\}$; $\mathcal{P} \leftarrow \text{Select}(\mathcal{H}, P)$
  **end for**
  **if** $r < R$ **then**
    Select top-$N$ coefficients $\{\boldsymbol{\lambda}^{(i)}\}_{i=1}^N$ from history $\mathcal{H}$
    Update experts: $\boldsymbol{\theta}_i \leftarrow \boldsymbol{\theta}(\boldsymbol{\lambda}^{(i)})$
    Re-compute task vectors: $\boldsymbol{\tau}_i \leftarrow \boldsymbol{\theta}_i - \boldsymbol{\theta}_{pre}$
  **end if**
**end for**
**return** best $\boldsymbol{\theta}(\boldsymbol{\lambda}^*)$ found in $\mathcal{H}$

---

**Population Initialization** The search process begins by initializing a population $\mathcal{P} = \{\boldsymbol{\lambda}^{(p)}\}_{p=1}^P$ of candidate configurations. Throughout this work, we set the population size to $P = 2N$. To ensure diverse coverage of the coefficient space, we employ a hybrid initialization strategy comprising: (i) average merging, where $\lambda_i = 1/N$ for all $i$; (ii) one-hot vectors, representing individual experts; and (iii) random sampling from a uniform distribution. Each candidate $\boldsymbol{\lambda}^{(p)}$ is evaluated on $\mathcal{D}_{val}$ to compute its performance score $f(\boldsymbol{\theta}(\boldsymbol{\lambda}^{(p)}), \mathcal{D}_{val})$. These initial evaluations establish the baseline performance and provide the essential seed trajectories for generative learning.

**Winner-Loser Pairing** In each iteration, we maintain a search history archive $\mathcal{H} = \{(\boldsymbol{\lambda}, f(\boldsymbol{\lambda}))\}$ that stores all previously evaluated coefficient configurations and their fitness scores. We split $\mathcal{H}$ into a winner set $\mathcal{H}^+$ and a loser set $\mathcal{H}^-$ using a ratio $\rho$ (default 0.3), where $\mathcal{H}^+$ contains the top-$\rho$ fraction of configurations. For clarity, we denote samples drawn from the loser and winner sets as $\boldsymbol{\lambda}^- \in \mathcal{H}^-$ and $\boldsymbol{\lambda}^+ \in \mathcal{H}^+$, respectively.

We construct a paired training set via the Cartesian product:

$$\mathcal{H}^- \times \mathcal{H}^+ = \{(\boldsymbol{\lambda}^-, \boldsymbol{\lambda}^+) \mid \boldsymbol{\lambda}^- \in \mathcal{H}^-, \boldsymbol{\lambda}^+ \in \mathcal{H}^+\}.$$

We form mini-batches by independently sampling $\boldsymbol{\lambda}^- \sim \mathcal{H}^-$ and $\boldsymbol{\lambda}^+ \sim \mathcal{H}^+$, which is equivalent to sampling from the product distribution.

**Train Dual-Generator**   To model transitions between different performance levels in the coefficient space $\mathbb{R}^N$, we employ a dual-generator architecture consisting of a forward generator $G_{-\to+}$ (mapping losers to winners) and a backward generator $G_{+\to-}$ (mapping winners to losers). Both generators are implemented as five-layer MLPs, with a `tanh` activation in the output layer to constrain the generated coefficients $\boldsymbol{\lambda}_{new}$ within the range $[-1, 1]$.

- **Training Objective.** The generators are jointly optimized by balancing cycle-consistency and optimization guidance:

$$\mathcal{L}_{total} = \alpha_c \, \mathcal{L}_{cyc} + \alpha_o \, \mathcal{L}_{opt},$$

where $\alpha_c$ and $\alpha_o$ control the contributions of cycle-consistency regularization and optimization guidance, respectively.

- **Cycle-Consistency Loss.** To ensure that the learned mapping is reversible and preserves the structural distribution of the coefficient population, we impose the following cycle-consistency constraint:

$$\mathcal{L}_{cyc} = \mathbb{E}_{\boldsymbol{\lambda}^- \in \mathcal{H}^-}\Big[\|G_{+\to-}(G_{-\to+}(\boldsymbol{\lambda}^-)) - \boldsymbol{\lambda}^-\|_2^2\Big]$$
$$+ \mathbb{E}_{\boldsymbol{\lambda}^+ \in \mathcal{H}^+}\Big[\|G_{-\to+}(G_{+\to-}(\boldsymbol{\lambda}^+)) - \boldsymbol{\lambda}^+\|_2^2\Big].$$

Minimizing $\mathcal{L}_{cyc}$ regularizes $G_{-\to+}$ to capture the geometric relationship between $\mathcal{H}^-$ and $\mathcal{H}^+$ and prevents trivial mode collapse.

- **Optimization-Guided Loss.** Beyond consistency, we explicitly guide the generative search toward high-performance regions. Let the centroid of the winner set be

$$\boldsymbol{\mu}^+ = \frac{1}{|\mathcal{H}^+|} \sum_{\boldsymbol{\lambda}^+ \in \mathcal{H}^+} \boldsymbol{\lambda}^+.$$

The optimization loss is defined as

$$\mathcal{L}_{opt} = \mathbb{E}_{\boldsymbol{\lambda}^- \in \mathcal{H}^-}\Big[\|G_{-\to+}(\boldsymbol{\lambda}^-) - \boldsymbol{\mu}^+\|_2^2\Big].$$

This objective drives $G_{-\to+}$ to transform sub-optimal configurations toward the statistical average of elite solutions, thereby accelerating the discovery of the global optimum $\boldsymbol{\lambda}^*$.

**Generative Evolution and Selection**   With the dual-generators optimized, the forward generator $G_{-\to+}$ serves as a learned evolutionary operator. In each iteration $t$, we generate a set of new candidates by applying the learned mapping to the current population $\mathcal{P}$: $\boldsymbol{\lambda}_{new} = G_{-\to+}(\boldsymbol{\lambda}), \quad \forall \boldsymbol{\lambda} \in \mathcal{P}$. By transforming the entire population through $G_{-\to+}$, the framework shifts the candidate

distribution toward regions with higher performance potential. These new candidates are subsequently evaluated on $\mathcal{D}_{val}$ to obtain their fitness $\mathcal{F}_{new}$. We then update the search history $\mathcal{H} \leftarrow \mathcal{H} \cup \{(\boldsymbol{\lambda}_{new}, \mathcal{F}_{new})\}$ and refine the population $\mathcal{P}$ by selecting the top-$P$ individuals based on their fitness scores. This process ensures that the population continuously migrates toward higher-performance regions within the coefficient space.

**Evolving Expert Basis**   A distinctive feature of EvoGM is that the expert pool itself evolves across outer rounds, which we refer to as a *basis shift*. While the inner loop optimizes the merging coefficients $\boldsymbol{\lambda}$ over a fixed set of task vectors, the outer loop explicitly redefines the underlying expert basis by updating the task vectors $\{\boldsymbol{\tau}_i\}$. At the end of each round ($r < R$), we select the top-$N$ coefficient vectors $\{\boldsymbol{\lambda}^{(i)}\}_{i=1}^N$ from the accumulated history archive $\mathcal{H}$. These elite configurations are then used to synthesize a new expert pool for the next round: $\boldsymbol{\theta}_i^{(r+1)} = \boldsymbol{\theta}(\boldsymbol{\lambda}^{(i)}), \quad i = 1, \ldots, N$, where $\boldsymbol{\theta}(\boldsymbol{\lambda}^{(i)})$ denotes the merged model constructed by applying $\boldsymbol{\lambda}^{(i)}$ to the experts of round $r$. We subsequently recompute task vectors with respect to the pretrained base model: $\boldsymbol{\tau}_i^{(r+1)} = \boldsymbol{\theta}_i^{(r+1)} - \boldsymbol{\theta}_{pre}$. This basis shift enables EvoGM to progressively refine the expert directions available for merging, rather than searching over a static expert pool.

## 3. Experimental Setup

We consider two model merging settings: merging for seen tasks and merging for unseen tasks. In this context, seen tasks refer to tasks on which the expert models are explicitly fine-tuned, while unseen tasks denote tasks that are not used during expert fine-tuning. Experiments span two model architectures, FLAN-T5 (Chung et al., 2024) and Qwen (Bai et al., 2023), different merging scales (8 and 10 models), and both single-task and multi-task configurations.

**Model Configuration**   For the seen tasks setting, we use FLAN-T5-base[1] together with 8 expert models fine-tuned on single corresponding tasks. For the unseen tasks setting, we construct a set of ten domain-specific experts[2] by independently fine-tuning Qwen2.5-1.5B [3] on each of the ten supervised fine-tuning (SFT) domains in Tulu-v2[4] (Ivison et al., 2023). All experts are adapted using LoRA-based parameter-efficient fine-tuning (Hu et al., 2022). Each model is trained for five epochs with an initial learning rate of $2 \times 10^{-4}$ and an effective batch size of 32.

---

[1] https://huggingface.co/google/flan-t5-base
[2] https://huggingface.co/TaoJiangCN/qwen2.5-1.5b-tulu-v2-lora-experts
[3] https://huggingface.co/Qwen/Qwen2.5-1.5B
[4] https://huggingface.co/collections/allenai/tulu-v2-suite

*Table 1.* Single-task performance comparison of 10 fine-tuned Qwen2.5-1.5B models. Best results are highlighted with **bold** and light-gray background, and second-best results are underlined, computed independently for each column (Val/Test).

| Method | MMLU | | MMLU-Pro | | HellaSwag | | K-Cross | | GSM8K | | NLGraph | | TruthQA | | AbstainQA | |
|---|---|---|---|---|---|---|---|---|---|---|---|---|---|---|---|---|
| | Val | Test | Val | Test | Val | Test | Val | Test | Val | Test | Val | Test | Val | Test | Val | Test |
| Base | 0.570 | 0.564 | 0.271 | 0.207 | 0.590 | 0.581 | 0.410 | 0.297 | 0.255 | 0.155 | 0.285 | 0.284 | 0.460 | 0.425 | 0.185 | 0.101 |
| MTL | 0.620 | 0.566 | 0.286 | 0.207 | 0.525 | 0.484 | 0.395 | 0.327 | 0.365 | 0.188 | 0.320 | 0.319 | 0.360 | 0.305 | 0.005 | 0.003 |
| Single Best | 0.615 | 0.586 | 0.386 | 0.232 | 0.620 | 0.572 | 0.410 | 0.392 | 0.395 | 0.325 | 0.355 | 0.376 | 0.495 | 0.384 | 0.210 | 0.119 |
| Model Soup (22) | 0.610 | 0.574 | 0.300 | 0.225 | 0.590 | 0.578 | 0.355 | 0.330 | 0.300 | 0.397 | 0.285 | 0.197 | 0.465 | 0.283 | 0.070 | 0.048 |
| TA (23) | 0.620 | 0.570 | 0.286 | 0.234 | 0.605 | 0.588 | 0.370 | 0.391 | 0.350 | 0.260 | 0.280 | 0.286 | 0.465 | 0.389 | 0.075 | 0.060 |
| TIES (23) | 0.600 | 0.557 | 0.329 | 0.221 | 0.510 | 0.470 | 0.345 | 0.342 | 0.295 | 0.302 | 0.390 | 0.316 | 0.495 | 0.416 | 0.075 | 0.028 |
| DARE (23) | 0.605 | 0.576 | 0.300 | 0.233 | 0.615 | 0.593 | 0.380 | 0.390 | 0.320 | 0.243 | 0.285 | 0.286 | 0.455 | 0.395 | 0.095 | 0.043 |
| CMA (25) | 0.615 | 0.576 | 0.386 | 0.217 | 0.600 | 0.592 | 0.415 | 0.392 | 0.430 | 0.343 | 0.430 | 0.445 | 0.530 | 0.428 | 0.170 | 0.075 |
| PSO-Merging (25) | 0.635 | 0.595 | 0.386 | 0.232 | 0.650 | 0.587 | 0.430 | 0.393 | 0.490 | 0.354 | 0.455 | 0.465 | 0.525 | 0.421 | 0.260 | 0.147 |
| Model Swarm (25) | 0.640 | 0.606 | 0.371 | 0.236 | 0.635 | 0.587 | 0.410 | 0.392 | 0.410 | 0.332 | 0.360 | 0.373 | 0.495 | 0.392 | 0.230 | 0.095 |
| Ours | 0.640 | 0.625 | 0.429 | 0.224 | 0.660 | 0.594 | 0.430 | 0.379 | 0.495 | 0.434 | 0.545 | 0.537 | 0.540 | 0.441 | 0.230 | 0.121 |

**Baselines** For the seen tasks setting, we compare against a broad range of representative merging approaches, including TA (Ilharco et al., 2023), DARE (Yu et al., 2024b), TIES (Yadav et al., 2023), DARE-TIES (Yadav et al., 2023; Yu et al., 2024b), DELLA-Merging (Deep et al., 2024), RankMean (Perin et al., 2024), CMA (Akiba et al., 2025), AdaMerging (Yang et al., 2024), Fisher-Merging (Matena & Raffel, 2022), and RegMean (Jin et al., 2023). For the unseen tasks setting, we consider both training-based and merging-based baselines. These include the base pretrained model (Base), multi-task learning (MTL), the best single expert (Single Best), and Model Soup (Wortsman et al., 2022), as well as merging methods such as TA, TIES, DARE, CMA, PSO-Merging (Zhang et al., 2025), and Model Swarm (Feng et al., 2025).

**Benchmarks and Tasks** For the seen tasks setting, we evaluate model merging on a suite of eight text-to-text generation tasks drawn from the GLUE benchmark (Wang et al., 2018), covering grammaticality judgment, natural language inference, paraphrase identification, sentiment analysis, and semantic similarity. The tasks include CoLA, MNLI, MRPC, QNLI, QQP, RTE, SST-2, and STS-B [5].

For the unseen tasks setting, we assess generalization on eight datasets spanning three categories: knowledge, reasoning, and safety. The knowledge category includes MMLU (Hendrycks et al., 2021), MMLU-Pro (Wang et al., 2024), and HellaSwag (Zellers et al., 2019); reasoning tasks cover GSM8K (Cobbe et al., 2021), Knowledge Crosswords (Ding et al., 2024), and NLGraph (Wang et al., 2023; Zhang et al., 2024); and safety-related evaluation is conducted on TruthfulQA (Lin et al., 2021), and Ab-

stainQA (Gehman et al., 2020). Unless otherwise specified, we randomly sample 200 examples for validation and 1,000 examples for testing on each dataset.

## 3.1. Seen Tasks

We evaluate the performance of merging 8 FLAN-T5 models on their corresponding tasks. Baseline results are taken from PSO-Merging for comparison. The multi-task performance on the GLUE benchmark is summarized in Table 2. Our proposed method consistently achieves the highest average accuracy among all tested approaches, providing a relative improvement of over 1.4% compared to the previous state-of-the-art, PSO-Merging. In semantic similarity (STS-B), our approach yields a relative performance gain of more than 12% over PSO-Merging and approximately 6% over the best-performing traditional merging baseline.

*Table 2.* Performance comparison on 8 GLUE benchmark tasks. Best results are highlighted with **bold** and light-gray background, and second-best results are underlined.

| Method | CoLA | MNLI | MRPC | QNLI | QQP | RTE | SST-2 | STS-B | AVG |
|---|---|---|---|---|---|---|---|---|---|
| TA | 69.1 | 62.7 | 79.4 | 89.8 | 83.9 | 81.2 | 91.7 | 73.2 | 78.9 |
| DARE | 69.5 | 63.8 | 79.7 | 89.9 | 83.9 | 81.2 | 91.7 | 69.8 | 78.7 |
| TIES | 69.2 | 59.4 | 77.7 | 89.3 | 83.4 | 80.5 | 91.3 | 68.4 | 77.4 |
| DARE-TIES | 69.3 | 62.5 | 79.7 | 89.8 | 83.8 | 81.6 | 91.3 | 71.1 | 78.6 |
| DELLA | 69.3 | 64.4 | 79.9 | 89.9 | 83.8 | 82.0 | 91.1 | 76.0 | 79.5 |
| RankMean | 69.1 | 56.5 | 76.2 | 88.5 | 82.1 | 80.1 | 91.2 | 62.2 | 75.7 |
| CMA | 70.9 | 82.9 | 75.7 | 89.4 | 73.9 | 80.9 | 92.2 | 69.7 | 79.4 |
| AdaMerging | 69.9 | 77.2 | 79.9 | 89.8 | 81.7 | 79.1 | 91.4 | 66.1 | 79.4 |
| Fisher | 69.3 | 54.0 | 76.7 | 84.6 | 83.6 | 77.6 | 88.1 | 74.4 | 76.0 |
| RegMean | 69.1 | 26.6 | 75.3 | 79.3 | 77.2 | 61.7 | 86.0 | 48.1 | 65.4 |
| PSO-Merging | 68.2 | 83.8 | 80.6 | 89.5 | 83.6 | 81.2 | 91.1 | 71.9 | 81.2 |
| Ours | 71.1 | 82.8 | 78.9 | 88.1 | 84.8 | 82.2 | 90.0 | 80.9 | 82.4 |

[5] https://huggingface.co/collections/tanganke/flan-t5-base-models-fine-tuned-on-glue-benchmark

*Table 3.* Multi-task performance comparison of 10 fine-tuned Qwen2.5-1.5B models. Best results are highlighted with **bold** and light-gray background, and second-best results are underlined, computed independently for each column (Val/Test).

| Method | MMLU | | MMLU-Pro | | HellaSwag | | K-Cross | | GSM8K | | NLGraph | | TruthQA | | AbstainQA | | Average |
|---|---|---|---|---|---|---|---|---|---|---|---|---|---|---|---|---|---|
| | Val | Test | Val | Test | Val | Test | Val | Test | Val | Test | Val | Test | Val | Test | Val | Test | |
| Base | 0.570 | 0.564 | 0.271 | 0.207 | 0.590 | 0.581 | 0.410 | 0.297 | 0.255 | 0.155 | 0.285 | 0.284 | 0.460 | 0.425 | 0.185 | 0.101 | 0.352 |
| MTL | 0.620 | 0.566 | 0.286 | 0.207 | 0.525 | 0.484 | 0.395 | 0.327 | 0.365 | 0.188 | 0.320 | 0.319 | 0.360 | 0.305 | 0.005 | 0.003 | 0.330 |
| Single Best | 0.615 | 0.586 | 0.386 | 0.232 | 0.620 | 0.572 | 0.410 | 0.392 | 0.395 | 0.325 | 0.355 | 0.376 | 0.495 | 0.384 | 0.210 | 0.119 | 0.405 |
| Model Soup (22) | 0.610 | 0.574 | 0.300 | 0.225 | 0.590 | 0.578 | 0.355 | 0.330 | 0.300 | 0.397 | 0.285 | 0.197 | 0.465 | 0.283 | 0.070 | 0.048 | 0.350 |
| Task Arithmetic (23) | 0.620 | 0.570 | 0.286 | 0.234 | 0.605 | 0.588 | 0.370 | 0.391 | 0.350 | 0.260 | 0.280 | 0.286 | 0.465 | 0.389 | 0.075 | 0.060 | 0.364 |
| TIES (23) | 0.600 | 0.557 | 0.329 | 0.221 | 0.510 | 0.470 | 0.345 | 0.342 | 0.295 | 0.302 | 0.390 | 0.316 | 0.495 | 0.416 | 0.075 | 0.028 | 0.356 |
| DARE (23) | 0.605 | 0.576 | 0.300 | 0.233 | 0.615 | 0.597 | 0.380 | 0.390 | 0.320 | 0.243 | 0.285 | 0.286 | 0.455 | 0.395 | 0.095 | 0.043 | 0.364 |
| CMA (25) | 0.600 | 0.569 | 0.357 | 0.217 | 0.645 | 0.609 | 0.405 | 0.415 | 0.275 | 0.211 | 0.275 | 0.275 | 0.500 | 0.421 | 0.140 | 0.078 | 0.375 |
| PSO-Merging (25) | 0.580 | 0.539 | 0.314 | 0.197 | 0.530 | 0.532 | 0.415 | 0.414 | 0.315 | 0.261 | 0.380 | 0.352 | 0.520 | 0.418 | 0.130 | 0.059 | 0.372 |
| Model Swarm (25) | 0.610 | 0.573 | 0.343 | 0.229 | 0.585 | 0.545 | 0.390 | 0.411 | 0.375 | 0.246 | 0.305 | 0.297 | 0.440 | 0.410 | 0.130 | 0.071 | 0.372 |
| Ours | 0.620 | 0.576 | 0.314 | 0.225 | 0.635 | 0.588 | 0.405 | 0.388 | 0.350 | 0.248 | 0.295 | 0.290 | 0.455 | 0.397 | 0.165 | 0.130 | 0.380 |

## 3.2. Unseen Tasks

Merging on unseen tasks effectively benchmarks generalization under data scarcity. By optimizing weights on small validation sets, evolutionary methods support both single-task specialization (tailoring weights to specific task) and multi-task merging (identifying a unified model for all tasks simultaneously).

**Single-task**  We evaluate the merging of 10 Qwen2.5-1.5B experts across 8 unseen tasks by optimizing dedicated weight configurations for each task independently. As shown in Table 1 and Figure 1, EvoGM achieves the highest test accuracy on 5 of 8 benchmarks. For instance, a 15% relative performance gain is observed in NLGraph compared to PSO-Merging. Notably, multi-task learning and traditional merging methods like TIES or DARE often fail to outperform the single best expert. This indicates that static merging schemes or joint training struggle to transfer specialized knowledge to new domains. In contrast, evolutionary methods such as CMA and PSO-Merging achieve better results by searching for optimal weights on validation sets. EvoGM further advances these baselines by incorporating learnable generative modeling, achieving superior and more robust performance across complex reasoning and knowledge tasks.

**Multi-task**  We evaluate the generalization of a single merged model across 8 unseen tasks simultaneously. As shown in Table 3, EvoGM achieves a leading average score of 0.380 among all merging approaches. Notably, standard multi-task fine-tuning yields a score of 0.330, which falls below the base model performance of 0.352. This performance drop underscores the severe inter-task interference inherent in joint fine-tuning across diverse domains. While evolutionary baselines such as CMA and PSO-Merging im-

prove results through weight search, their efficacy is often constrained by the stochasticity of their operators. Conversely, EvoGM navigates the high-performance parameter space more effectively through its generative architecture. This advantage is especially pronounced in AbstainQA, where EvoGM exhibits a 66% relative improvement over the strongest evolutionary baseline. These results demonstrate that by leveraging historical search trajectories, our framework discovers balanced weight combinations that successfully mitigate task conflicts in few-shot scenarios.

## 4. Analysis

**Convergence and Optimization Efficiency**  We analyze the convergence of EvoGM by comparing it with Model Swarm and PSO-Merging. All methods are evaluated using a population size of 20. To visualize the optimization process, we plot the average performance of the top 5 solutions from each iteration.

As shown in Figure 3, EvoGM consistently outperforms the baselines across all tasks. The baseline algorithms typically reach a performance plateau around iteration 5, showing little improvement thereafter. In contrast, EvoGM follows a multi-round optimization strategy, configured here with 2 rounds and 3 iterations per round. A noticeable performance shift occurs at iteration 3, which marks the transition between rounds. At this stage, the framework performs a basis shift by generating new parameter candidates based on the elite models from the previous round. While this re-initialization causes a temporary fluctuation, it effectively prevents the search from getting trapped in local optima. This phase allows EvoGM to explore more promising regions of the parameter space, leading to substantial performance gains in the subsequent round.

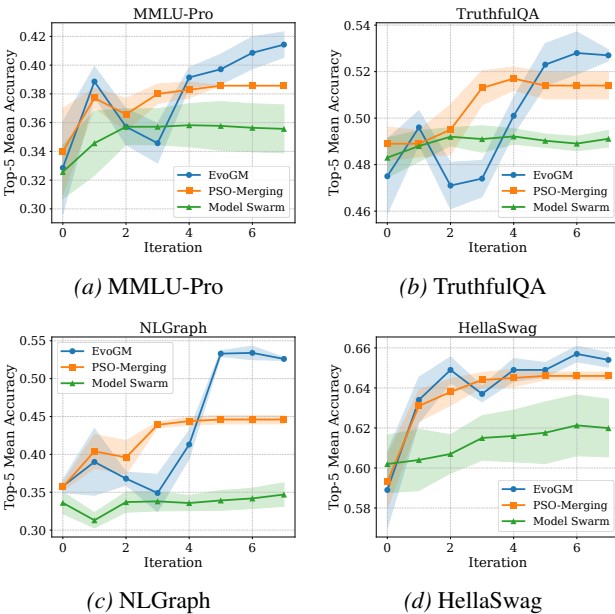

*Figure 3.* Fitness evolution of EvoGM and SOTA methods in multi-task scenarios. The curves represent the mean performance of the top five individuals in the population, with shaded areas indicating the confidence intervals computed from these individuals.

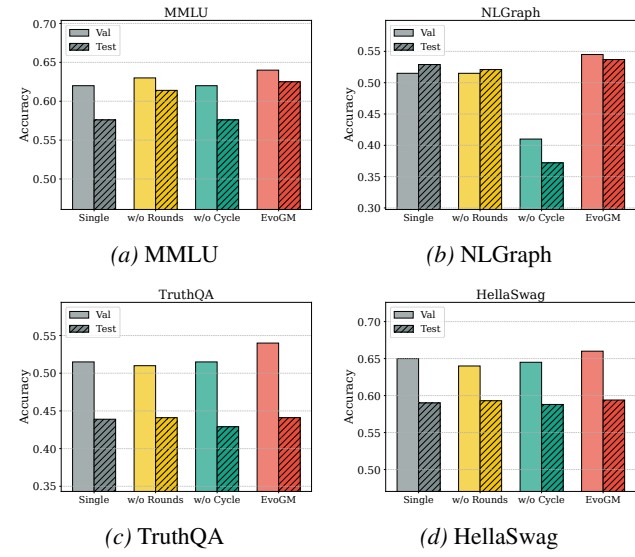

*Figure 4.* Ablation study on different components. (1) Single-Generator represents the single generative model variant; (2) w/o Rounds represents the replacement of multi-round updates with five continuous iterations; and (3) w/o Cycle Loss represents the removal of the cycle-consistency constraint.

**Ablation Study**  We conduct an ablation study to evaluate the contribution of each core component in EvoGM. This experiment involves merging 4 expert models with a population size of 8. The full framework is configured with 2 rounds and 2 iterations per round. We compare the full version against 3 variants: (1) Single-Generator, which uses only a single generative model; (2) w/o Rounds, which replaces the multi-round expert update with 5 continuous iterations on the initial experts; and (3) w/o Cycle Loss, which removes the cycle-consistency constraint. Results are summarized in Figure 4 and Table 11 in Appendix.

Each component is shown to be critical for achieving optimal performance across diverse benchmarks. The removal of the multi-round mechanism (w/o Rounds) leads to a noticeable decline in complex tasks such as NLGraph and MMLU. This confirms that iteratively updating the expert foundations allows the model to escape local optima and integrate knowledge more deeply. Furthermore, the variant without cycle loss (w/o Cycle Loss) suffers a significant performance drop, particularly in reasoning tasks like NLGraph. This suggests that the cycle-consistent constraint is vital for accurately capturing the high-performance regions of the parameter space. Finally, the dual-generator architecture consistently outperforms the single-generator variant, proving that bidirectional learning of winner-loser pairs more effectively identifies optimal merging coefficients.

**Hyperparameter Sensitivity**  We evaluate the sensitivity of EvoGM to key hyperparameters by merging 3 models

and reporting the average performance across 2 tasks. As summarized in Table 12 of the Appendix, the population size exhibits a significant influence on search quality. Increasing the population from 10 to 30 leads to a 0.0305 gain in test accuracy. This suggests that a larger number of candidates allows for a more thorough exploration of the high-dimensional parameter manifold. In contrast, the number of iterations per round has a relatively stable effect, with a slight performance increase observed as iterations scale from 3 to 8.

Regarding the training of the generative model, we find that 200 to 400 epochs are sufficient. Increasing the epochs to 600 causes a minor performance drop of 0.0035, likely due to the generator over-fitting on specific historical search trajectories. For the learning rate, a smaller value of 0.0001 yields the best results, providing a 0.0220 improvement over the baseline. Finally, increasing the evolution rounds from 1 to 5 consistently enhances performance. This confirms that the iterative multi-round process is essential for refining expert foundations and achieving higher generalization.

**Impact of Model Scale**  We investigate the scalability of EvoGM by varying the number of merged experts from 2 to 10 in a multi-task setting. Results are summarized in Figure 5 and Table 13 in Appendix, where we compare the initial performance against the final optimized results.

Our framework consistently improves average accuracy across all model counts. When merging 8 models, the average test score increases from 0.433 to 0.460. This trend

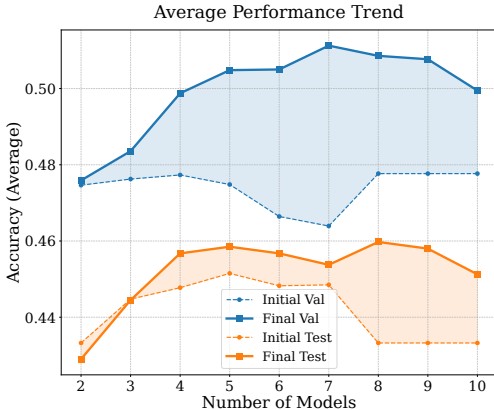

*Figure 5.* Impact of the number of merged models on performance. We evaluate how the merging quality scales when integrating different quantities of models.

indicates that EvoGM is highly robust to the initial configuration and the complexity of the weight space. Even as the number of experts increases, the generative evolutionary process successfully identifies superior merging coefficients that mitigate task interference. These results demonstrate that our method scales effectively, maintaining consistent performance gains regardless of whether a small ensemble or a larger set of 10 experts is used. This stability proves that EvoGM effectively navigates the high-dimensional parameter space across various scales of model composition.

**Extended Scaling and Robustness Experiments** We further examine EvoGM under larger model and expert-space settings, with full results reported in Appendix B. To test model-scale transfer, we merge ten Tulu-finetuned experts built on Qwen3-8B; EvoGM achieves the best average score among all compared methods (0.603) while maintaining competitive wall-clock search cost. To test scalability with respect to the number of experts, we additionally evaluate a 20-expert ViT-B-16 merging setting under the same validation/test split and search budget as CMA-ES, where EvoGM obtains a higher average test score (0.6363 vs. 0.6268). Finally, multi-seed experiments on Qwen2.5-1.5B show statistically significant average improvements over PSO-Merging ($p = 0.0100$) and Model Swarm ($p = 0.0006$), indicating that the performance gains are not due to random variation.

## 5. Related Work

**Model Merging** Model merging facilitates the integration of diverse expert models into a unified framework without additional training costs (Yang et al., 2026). Early research demonstrates that models fine-tuned from the same initialization can be linearly combined when they reside in compatible loss basins (Ilharco et al., 2023; Wortsman et al., 2022). However, direct weight averaging often suffers from parame-

ter interference. To address this, subsequent studies propose conflict-aware mechanisms, such as sign-based trimming (TIES) (Yadav et al., 2023), importance-aware pruning and rescaling (DU et al., 2024), and the selective removal of redundant or conflicting components (Sun et al., 2025a). More recently, these techniques have been extended to Parameter-Efficient Fine-Tuning (PEFT) and multimodal settings (Li et al., 2025c), including LoRA-style merging (Panariello et al., 2025) and dynamic routing frameworks for heterogeneous architectures (Chen et al., 2025; Lu et al., 2024b).

Beyond direct parameter manipulation, another research thrust focuses on improving model mergeability through representation-level synchronization. This includes resolving neuron permutation symmetries (Crisostomi et al., 2024), enforcing dual-space consistency (Xu et al., 2024b), and performing subspace alignment or activation-level corrections (Horoi et al., 2024; Shao et al., 2026; Sun et al., 2025c; Yao et al., 2025). To further enhance robustness across heterogeneous task distributions, iterative merging strategies have been proposed to refine weights through multi-stage compositions (Tang et al., 2025; Yuan et al., 2025).

While early methods rely on hand-crafted heuristics, recent empirical evidence suggests that performance is often governed by the precision of merging coefficients rather than the specific operators used (Lan et al., 2025). This shift in perspective has motivated the use of formal optimization to determine optimal blending weights. Current approaches employ entropy-based objectives (Yang et al., 2024), Bayesian inference (Li et al., 2025b), and Pareto-aware multi-objective designs (Chen & Kwok, 2025; Li et al., 2024; Zhou et al., 2025) to explore the weight space. However, these methods often rely on stochastic search or static probabilistic priors, leaving a gap for more adaptive, generative search mechanisms that can effectively capture high-performance parameter configurations.

**Evolutionary Model Merging** A burgeoning line of research reframes model merging as a gradient-free optimization challenge, moving away from manual heuristic design. Evolutionary Algorithms (EAs) have become a preferred paradigm due to their flexibility in handling non-differentiable objectives. One category focuses on optimizing merging hyperparameters, such as layer-wise coefficients and weight sparsity ratios (Akiba et al., 2025; Baba et al., 2024). Alternatively, population-centric methods evolve the models themselves, employing mutation and crossover operators to generate novel architectural candidates (Du et al., 2024). More recent swarm-intelligence frameworks harness collaborative search trajectories to identify optimal weight configurations in the parameter space (Feng et al., 2025; Zhang et al., 2025).

Despite their potential, evolutionary model merging methods are often bottlenecked by the prohibitive computational cost of evaluating large-scale models. Scalable evolutionary-computation platforms such as EvoX (Huang et al., 2024) have improved the practicality and programmability of evolutionary search, and efficiency-oriented merging frameworks further mitigate evaluation costs through surrogate models or sparse validation subsets (Akizuki et al., 2025; Mencattini et al., 2025). Yet these advances mainly concern the execution or evaluation of search, leaving the generation of candidate merging configurations largely governed by generic stochastic operators. These operators often lack the structural awareness needed to navigate high-dimensional merging landscapes, leading to poor sample efficiency under limited evaluation budgets.

## 6. Conclusion and Limitations

**Conclusion.** This paper presents EvoGM, a framework that reformulates model merging as a learnable generative task. By integrating dual-generator learning with a hierarchical basis shift mechanism, EvoGM effectively navigates the complex performance landscape of large language models without relying on manual heuristics. Our experiments demonstrate that by learning from historical search trajectories, EvoGM achieves superior data efficiency and discovers high-synergy model configurations that outperform current state-of-the-art baselines across various benchmarks.

**Limitations.** The current study focuses on homogeneous task-vector merging, where all expert models are obtained from the same pretrained base model. This setting makes the linear coefficient space well-defined, but it also limits the direct use of EvoGM in more general merging scenarios. For models with different architectures, initializations, or parameter structures, additional alignment is needed before coefficient-based search can be applied. Moreover, our multi-task setting optimizes a single scalar validation objective and returns one deployable merged model. This is suitable when a fixed task balance is given, but it does not directly provide a set of Pareto-optimal models for users with different preferences. Extending EvoGM to heterogeneous model merging and preference-aware multi-objective merging is therefore an important direction for future work.

## Acknowledgements

This work is supported in part by National Natural Science Foundation of China (Grant No. 62276121), in part by the TianYuan funds for Mathematics of the National Science Foundation of China (Grant No. 12326604), in part by Innovation Team and Talents Cultivation Program of National Administration of Traditional Chinese Medicine (No: ZYYCXTD-D-202403), in part by Guangdong Basic and Applied Basic Research Foundation (No. 2024B1515020019), in part by National Natural Science Foundation of China (Grant No. 62502246).

## Impact Statement

This work studies training-free model merging for composing multiple fine-tuned models. By learning to propose high-performing merge coefficients from historical search trajectories, EvoGM can reduce the need to retrain full models or maintain many separate expert models. This makes model adaptation more practical when compute, validation data, or engineering resources are limited. More broadly, efficient model merging can encourage the reuse of existing task-specialized checkpoints, improve the accessibility of customized AI systems, and support more sustainable model development by amortizing prior fine-tuning costs across multiple downstream tasks.

The societal effects of EvoGM are therefore expected to be largely aligned with those of efficient model reuse and composition. As with other model-merging methods, a merged model may inherit biases, privacy concerns, or unsafe behaviors from its source models, so appropriate source-model selection and task-relevant evaluation remain important.

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

## A. Complexity Analysis

To evaluate the efficiency of EvoGM, we analyze its computational overhead in terms of both time and space complexity. Let $N$ denote the number of expert models, $D$ the number of parameters per model (or LoRA rank), $R$ the number of outer rounds, $T$ the inner evolutionary iterations, $P$ the population size, and $E$ the cost of a single evaluation on the validation set.

**Time Complexity.** The time complexity of EvoGM is primarily dominated by the iterative evaluation and expert refinement phases. Initially, computing task vectors requires $O(ND)$ operations in the parameter space. During the main loop, the generative optimization occurs in the low-dimensional coefficient space with a negligible cost of $O(RTP \cdot \mathrm{poly}(N))$, as $N \ll D$. The most significant bottleneck is the fitness evaluation of candidate models, which sequentially takes $O(RTPE)$. However, since evaluations of candidate coefficients are embarrassingly parallel, the total wall-clock time can be significantly reduced by employing $M$ parallel workers. In such a parallelized setting, the time complexity is optimized to $O(ND + RT\lceil P/M \rceil E + RKND)$, where the final term accounts for the $K$ model merging operations per round during expert refinement. This linear scalability with respect to $M$ allows EvoGM to handle large populations and intensive search rounds efficiently.

**Space Complexity.** In terms of space complexity, EvoGM is designed to be highly memory-efficient, especially within parameter-efficient fine-tuning (PEFT) frameworks. The primary memory footprint consists of the frozen backbone model and $N$ task vectors, resulting in a total requirement of $O(D_{base} + ND_{task})$. Throughout the evolutionary process, the algorithm only needs to store a history of population coefficients and their corresponding scores, which occupies $O(RTPN)$ space. Given that $N$ is typically small, this storage cost is marginal compared to the model parameters. During the evaluation phase, EvoGM avoids the need for multiple full-model replicas by dynamically merging task vectors in-place, maintaining a consistent space complexity of $O(D + ND)$. This efficiency ensures that our approach remains feasible even for large-scale language models with limited hardware resources.

## B. Additional Robustness and Scaling Results

For the experiments reported in this section, we used Ascend 910 NPUs together with HiSilicon AArch64 processors. The other experiments were conducted on NVIDIA A100-SXM4-40GB GPUs with AMD EPYC 7742 64-Core processors. Since these platforms differ in their underlying hardware architectures, minor discrepancies in absolute performance may arise. However, within each experimental setting, our method and all baselines were executed on identical hardware, thereby ensuring the fairness of the comparative evaluation.

### B.1. Statistical Robustness

To assess whether small average gains are driven by random variation, we reran EvoGM, PSO-Merging, and Model Swarm with matched random seeds on the Qwen2.5-1.5B unseen-task benchmark. Table 4 reports mean and standard deviation over four runs and independent two-sample Welch's t-tests. While not every task-level difference is significant, EvoGM obtains a statistically significant improvement in the average score over PSO-Merging ($p = 0.0100$) and Model Swarm ($p = 0.0006$).

*Table 4.* Multi-seed statistical robustness on Qwen2.5-1.5B unseen tasks.

| Task | EvoGM mean ± std | PSO-Merging mean ± std | Model Swarm mean ± std | EvoGM vs. PSO $p$-value | EvoGM vs. Swarm $p$-value |
|---|---|---|---|---|---|
| MMLU | $0.5942 \pm 0.0184$ | $0.5990 \pm 0.0072$ | $0.6015 \pm 0.0163$ | 0.6614 | 0.5770 |
| MMLU-Pro | $0.2412 \pm 0.0084$ | $0.2298 \pm 0.0072$ | $0.2400 \pm 0.0139$ | 0.0840 | 0.8839 |
| HellaSwag | $0.5930 \pm 0.0074$ | $0.5925 \pm 0.0115$ | $0.5900 \pm 0.0262$ | 0.9446 | 0.8377 |
| Knowledge Crosswords | $0.4080 \pm 0.0041$ | $0.3853 \pm 0.0418$ | $0.3948 \pm 0.0068$ | 0.3563 | 0.0211 |
| GSM8K | $0.4408 \pm 0.0256$ | $0.3675 \pm 0.0454$ | $0.3295 \pm 0.0070$ | 0.0400 | 0.0021 |
| NLGraph | $0.5262 \pm 0.0184$ | $0.5443 \pm 0.0506$ | $0.3748 \pm 0.0005$ | 0.5420 | 0.0005 |
| TruthfulQA | $0.4344 \pm 0.0159$ | $0.4315 \pm 0.0184$ | $0.4303 \pm 0.0148$ | 0.8234 | 0.7220 |
| MMLU-Abstain | $0.1333 \pm 0.0223$ | $0.1137 \pm 0.0172$ | $0.1330 \pm 0.0265$ | 0.2190 | 0.9890 |
| **Average** | $0.4214 \pm 0.0073$ | $0.4018 \pm 0.0019$ | $0.3867 \pm 0.0076$ | **0.0100** | **0.0006** |

## B.2. Fairness of the Basis Shift

To test whether EvoGM benefits merely from being granted a multi-round basis shift, we also applied the same two-round, three-iteration-per-round schedule to PSO-Merging. As shown in Table 5, simply adding the same basis-reset mechanism to PSO-Merging reduces the average score from 0.4018 to 0.3425. This indicates that the basis shift is not a generic advantage for any search method; in EvoGM, it works together with learned winner-loser mappings and cycle consistency to preserve useful search structure across rounds.

*Table 5.* Single-round PSO, multi-round PSO with the same basis-reset schedule, and multi-round EvoGM on Qwen2.5-1.5B.

| Task | PSO Single-Round | PSO Multi-Round | EvoGM Multi-Round |
|---|---|---|---|
| MMLU | 0.5990 | 0.5660 | 0.5942 |
| MMLU-Pro | 0.2298 | 0.2070 | 0.2412 |
| HellaSwag | 0.5925 | 0.5840 | 0.5930 |
| Knowledge Crosswords | 0.3853 | 0.3790 | 0.4080 |
| GSM8K | 0.3675 | 0.2060 | 0.4408 |
| NLGraph | 0.5443 | 0.2820 | 0.5262 |
| TruthfulQA | 0.4315 | 0.4279 | 0.4344 |
| MMLU-Abstain | 0.1137 | 0.0880 | 0.1333 |
| **Average** | **0.4018** | **0.3425** | **0.4214** |

## B.3. Scaling and Search Cost on Qwen3-8B

We further evaluate EvoGM on Qwen3-8B with ten Tulu-finetuned experts. Table 6 shows that EvoGM achieves the best average score among the tested methods. We also report wall-clock time and forward-pass counts in Table 7. Although EvoGM uses more candidate evaluations than Model Swarm in this setup, the total wall-clock time remains competitive because candidate evaluation is parallelizable and the small MLP generators add negligible overhead relative to model inference.

*Table 6.* Qwen3-8B test performance with ten Tulu-finetuned experts.

| Method | MMLU | MMLU-Pro | HellaSwag | K-Cross | GSM8K | NLGraph | TruthQA | AbstainQA | AVG |
|---|---|---|---|---|---|---|---|---|---|
| Base | 0.655 | 0.349 | 0.792 | 0.589 | 0.125 | 0.534 | 0.561 | 0.308 | 0.489 |
| MTL | 0.702 | 0.399 | 0.712 | 0.667 | 0.428 | 0.363 | 0.558 | 0.296 | 0.516 |
| Single Best | **0.732** | **0.432** | 0.802 | 0.644 | 0.614 | 0.548 | 0.630 | **0.400** | 0.600 |
| Task Arithmetic | 0.721 | **0.432** | 0.809 | 0.580 | 0.355 | 0.413 | 0.566 | 0.232 | 0.513 |
| TIES Merging | 0.673 | 0.369 | 0.805 | 0.558 | 0.214 | 0.457 | 0.571 | 0.349 | 0.500 |
| Model Swarm | 0.712 | 0.420 | 0.789 | **0.669** | 0.588 | 0.551 | **0.637** | 0.381 | 0.593 |
| **EvoGM** | 0.718 | 0.429 | **0.824** | 0.586 | **0.655** | **0.607** | 0.614 | 0.394 | **0.603** |

*Table 7.* Search cost on Qwen3-8B using the Ascend 910 NPUs together with HiSilicon AArch64 processors. FPs denotes candidate forward-pass evaluations on validation data.

| Task | EvoGM Time | EvoGM FPs | Model Swarm Time | Model Swarm FPs |
|---|---|---|---|---|
| MMLU | 43.1 min | 160 | 32.5 min | 100 |
| MMLU-Pro | 40.4 min | 160 | 26.2 min | 100 |
| HellaSwag | 38.4 min | 160 | 31.0 min | 120 |
| K-Cross | 37.7 min | 160 | 40.0 min | 120 |
| GSM8K | 86.5 min | 160 | 98.2 min | 100 |
| NLGraph | 58.5 min | 160 | 68.3 min | 100 |
| TruthQA | 38.7 min | 160 | 23.0 min | 100 |
| AbstainQA | 42.2 min | 160 | 109.7 min | 160 |
| **Total** | **385.5 min** | **1280** | **428.9 min** | **900** |

## B.4. Higher-Dimensional Expert Space

To probe a larger coefficient space, we compare EvoGM with CMA-ES on a ViT-B-16 model merging benchmark with 20 experts. Both methods use the same validation/test subsets and the same six-iteration search budget. Table 8 shows

that EvoGM obtains a higher average test score than CMA-ES (0.6363 vs. 0.6268). It suggests that the generative search mechanism remains useful beyond the 8-10 expert settings used in the main experiments.

*Table 8.* ViT-B-16 merging with 20 experts under an identical six-iteration budget.

| Task | CMA-ES | EvoGM |
|------|--------|-------|
| Cars | 0.5028 | **0.7006** |
| DTD | **0.6528** | 0.5111 |
| EuroSAT | 0.4828 | **0.8106** |
| GTSRB | **0.7794** | 0.5061 |
| MNIST | **0.7050** | 0.5894 |
| RESISC45 | 0.6767 | **0.7367** |
| SVHN | **0.8956** | 0.4556 |
| SUN397 | 0.6206 | **0.6783** |
| STL10 | 0.9622 | **0.9783** |
| OxfordIIITPet | **0.9189** | 0.8439 |
| Flowers102 | 0.6172 | **0.6217** |
| CIFAR100 | 0.7300 | **0.7611** |
| PCAM | **0.5744** | 0.5706 |
| FER2013 | **0.4922** | 0.4394 |
| CIFAR10 | 0.8511 | **0.9433** |
| Food101 | 0.6406 | **0.8867** |
| FashionMNIST | 0.5128 | **0.8044** |
| RenderedSST2 | 0.4872 | **0.6433** |
| EMNIST | **0.2939** | 0.1294 |
| KMNIST | **0.1406** | 0.1161 |
| **Average** | 0.6268 | **0.6363** |

# C. Experimental Details

## C.1. Baselines

We select a total of 13 representative model merging methods as baselines. The details of these methods are summarized below.

- **Model Soup:** This approach merges multiple fine-tuned models by uniformly averaging their parameters (Wortsman et al., 2022). It assumes that independently trained models lie in a shared low-loss basin, enabling effective aggregation without additional tuning.

- **TA (Task Arithmetic):** Task Arithmetic (Ilharco et al., 2023) represents task-specific adaptations as parameter difference vectors relative to a base model and combines them through linear addition. This formulation enables compositional transfer across tasks but may suffer from interference under naive aggregation.

- **DARE:** DARE mitigates parameter conflicts by sparsifying task vectors before aggregation (Yu et al., 2024b). By retaining only high-magnitude parameter updates, it reduces destructive interference during merging.

- **TIES:** Rather than averaging all parameters, this method resolves sign conflicts across task vectors and selectively trims inconsistent dimensions (Yadav et al., 2023). The resulting merge emphasizes parameters with consistent directional contributions.

- **DARE-TIES:** Combining sparsification and sign-based conflict resolution, this variant first applies DARE-style pruning and then performs TIES-based trimming (Yadav et al., 2023; Yu et al., 2024b). The hybrid design aims to further suppress interference while preserving task-relevant updates.

- **DELLA:** This method identifies task-relevant subspaces through low-rank decomposition and performs merging within these aligned representations (Deep et al., 2024). By operating in a shared latent space, it improves compatibility across merged models.

- **RankMean:** Instead of averaging raw parameter values, RankMean aggregates parameters based on their relative rankings across models (Perin et al., 2024). This rank-based formulation increases robustness to scale mismatches and outlier updates.

- **CMA:** CMA formulates model merging as a black-box optimization problem and applies covariance matrix adaptation to search for optimal merging coefficients (Akiba et al., 2025). The method iteratively updates a population of candidates based on validation performance.

- **AdaMerging:** This approach learns adaptive, layer-wise merging coefficients guided by validation feedback (Yang et al., 2024). The coefficients are optimized to balance task contributions across different network depths.

- **Fisher:** Fisher-weighted merging leverages Fisher information to estimate parameter importance for each model (Matena & Raffel, 2022). Parameters deemed more critical receive higher weights during aggregation, reducing harmful interference.

- **RegMean:** By incorporating regularization terms during aggregation, this method penalizes deviations from important parameters of individual models (Jin et al., 2023). The regularized formulation stabilizes merging under heterogeneous tasks.

- **PSO-Merging:** PSO-Merging formulates model merging as a particle swarm optimization problem in parameter space (Zhang et al., 2025). It iteratively updates candidate merged models based on personal and global best solutions guided by task performance, enabling data-driven yet gradient-free merging.

- **Model Swarm:** Model Swarm treats each expert model as a particle and performs collaborative search in the weight space under a task-specific utility function (Feng et al., 2025). Instead of static aggregation, it adaptively explores model combinations and returns the best-performing expert discovered during optimization.

*Table 9.* Details of task datasets used in our experiments.

| Dataset | Source | Size | | Domain |
|---------|--------|------|------|--------|
| | | Val | Test | |
| CoLA | (Warstadt et al., 2019) | 50 | 1043 | Misc. |
| MNLI | (Williams et al., 2018) | 50 | 9815 | Misc. |
| MRPC | (Dolan & Brockett, 2005) | 50 | 408 | News |
| QNLI | (Levesque et al., 2011) | 50 | 5463 | Wikipedia |
| QQP[6] | / | 50 | 40430 | Social QA questions |
| RTE | (Bar-Haim et al., 2006; Bentivogli et al., 2009; Dagan et al., 2006; Giampiccolo et al., 2007) | 50 | 277 | News, Wikipedia |
| SST-2 | (Socher et al., 2013) | 50 | 872 | Movie reviews |
| STS-B | (Cer et al., 2017) | 50 | 1500 | Misc. |
| MMLU | (Hendrycks et al., 2021) | 200 | 1000 | Knowledge |
| MMLU-Pro | (Wang et al., 2024) | 70 | 1000 | Knowledge |
| Hellaswag | (Zellers et al., 2019) | 200 | 1000 | Knowledge |
| K-Cross | (Ding et al., 2024) | 200 | 1000 | Cross-domain reasoning |
| GSM8k | (Cobbe et al., 2021) | 200 | 1000 | Reasoning tasks |
| NLGraph | (Wang et al., 2023; Zhang et al., 2024) | 200 | 1000 | Reasoning tasks |
| TruthfulQA | (Lin et al., 2021) | 200 | 617 | Safety-related evaluation |
| AbstainQA | (Gehman et al., 2020) | 200 | 1000 | Safety-related evaluation |

## C.2. Datasets

We use sixteen datasets from established benchmarks, covering natural language understanding, reasoning, and safety evaluation. Eight datasets are drawn from the GLUE benchmark (Wang et al., 2018), including CoLA, MNLI, MRPC, QNLI, QQP, RTE, SST-2, and STS-B, which together represent a broad spectrum of natural language understanding tasks. In addition, we include eight widely used datasets for evaluating knowledge, reasoning, and safety evaluation, which are MMLU (Hendrycks et al., 2021), MMLU-Pro (Wang et al., 2024), HellaSwag (Zellers et al., 2019), GSM8K (Cobbe et al., 2021), K-Cross (K-Cross) (Ding et al., 2024), NLGraph (Wang et al., 2023; Zhang et al., 2024), TruthfulQA (Lin et al., 2021), and AbstainQA (Gehman et al., 2020). Detailed dataset statistics are reported in Table 9.

## C.3. Expert Models

Our experiments involve a total of eighteen expert models to be merged. All experts share the same architecture within each setting and are independently adapted from a common base model using different training datasets. An overview of the expert models and their corresponding training data is provided in Table 10. Eight experts are obtained by fine-tuning FLAN-T5-base[7] on individual GLUE datasets (Wang et al., 2018), while the remaining ten experts are adapted from Qwen2.5-1.5B[8] using LoRA-based parameter-efficient fine-tuning on supervised fine-tuning datasets from the Tulu-v2[9] (Ivison et al., 2023) collection. All experts are trained for five epochs with an initial learning rate of $2 \times 10^{-4}$ and an effective batch size of 32.

*Table 10.* Expert models used in our experiments.

| Base model | Source | Train Dataset | Models |
|---|---|---|---|
| FLAN-T5[10] | Open models | CoLA (Warstadt et al., 2019) | FLAN-T5-cola[11] |
| | | MNLI (Williams et al., 2018) | FLAN-T5-mnli[12] |
| | | MRPC (Dolan & Brockett, 2005) | FLAN-T5-mrpc[13] |
| | | QNLI (Levesque et al., 2011) | FLAN-T5-qnli[14] |
| | | QQP[15] | FLAN-T5-qqp[16] |
| | | RTE (Bar-Haim et al., 2006; Bentivogli et al., 2009; Dagan et al., 2006; Giampiccolo et al., 2007) | FLAN-T5-rte[17] |
| | | SST-2 (Socher et al., 2013) | FLAN-T5-sst2[18] |
| | | STS-B (Cer et al., 2017) | FLAN-T5-stsb[19] |
| QWEN2.5 -1.5B[20] | LoRA-based fine-tuning | flan (Chung et al., 2024) | - |
| | | CoT (Wei et al., 2022) | - |
| | | OpenAssistant-1 (Köpf et al., 2023) | - |
| | | ShareGPT [21] | - |
| | | Code Alpaca (Chaudhary, 2023) | - |
| | | LIMA (Zhou et al., 2023) | - |
| | | WizardLM Evol-Instruct v2 (Xu et al., 2024a) | - |
| | | Open-Orca (Lian et al., 2023) | - |
| | | Science Literature (Ivison et al., 2023) | - |
| | | Gemini Alpaca (distilled) | - |

## C.4. More Experimental Results

*Table 11.* Ablation study of EvoGM on 8 single-task benchmarks. Best results are highlighted with **bold** and a light-gray background, computed independently for each column (Val/Test).

| Method | MMLU | | MMLU-Pro | | HellaSwag | | K-Cross | | GSM8K | | NLGraph | | TruthQA | | AbstainQA | |
|---|---|---|---|---|---|---|---|---|---|---|---|---|---|---|---|---|
| | Val | Test | Val | Test | Val | Test | Val | Test | Val | Test | Val | Test | Val | Test | Val | Test |
| Single-Generator | 0.620 | 0.576 | 0.385 | 0.232 | 0.650 | 0.590 | 0.425 | 0.412 | **0.540** | 0.339 | 0.515 | 0.529 | 0.515 | 0.439 | 0.210 | 0.119 |
| w/o Rounds | 0.630 | 0.614 | 0.400 | **0.247** | 0.640 | 0.593 | **0.450** | **0.417** | 0.510 | 0.337 | 0.515 | 0.521 | 0.510 | **0.441** | 0.215 | **0.142** |
| w/o Cycle Loss | 0.620 | 0.576 | 0.386 | 0.232 | 0.645 | 0.588 | 0.435 | 0.396 | 0.410 | 0.326 | 0.410 | 0.372 | 0.515 | 0.429 | 0.210 | 0.119 |
| EvoGM | **0.640** | **0.625** | **0.429** | 0.224 | **0.660** | **0.594** | **0.450** | 0.379 | 0.495 | **0.434** | **0.545** | **0.537** | **0.540** | **0.441** | **0.230** | 0.121 |

---

[7] https://huggingface.co/google/flan-t5-base
[8] https://huggingface.co/Qwen/Qwen2.5-1.5B
[9] https://huggingface.co/collections/allenai/tulu-v2-suite

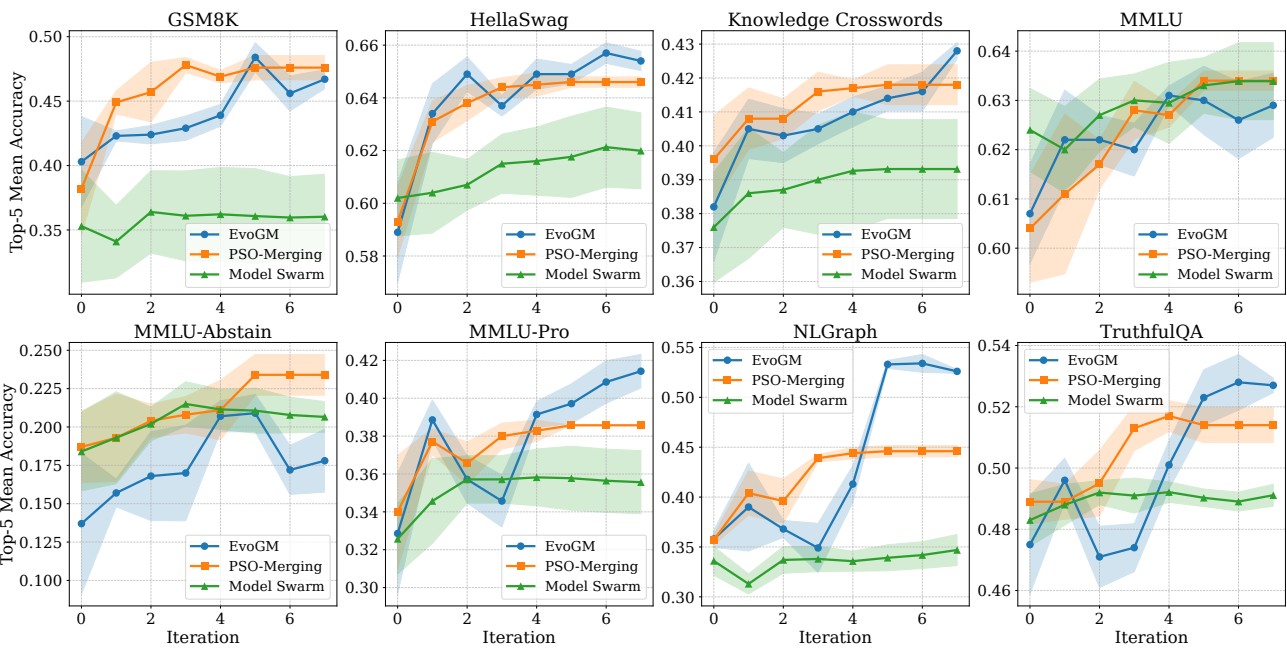

*Figure 6.* Fitness evolution of EvoGM and SOTA methods in multi-task scenarios. The curves represent the mean performance of the top five individuals in the population, with shaded areas indicating the confidence intervals computed from these individuals.

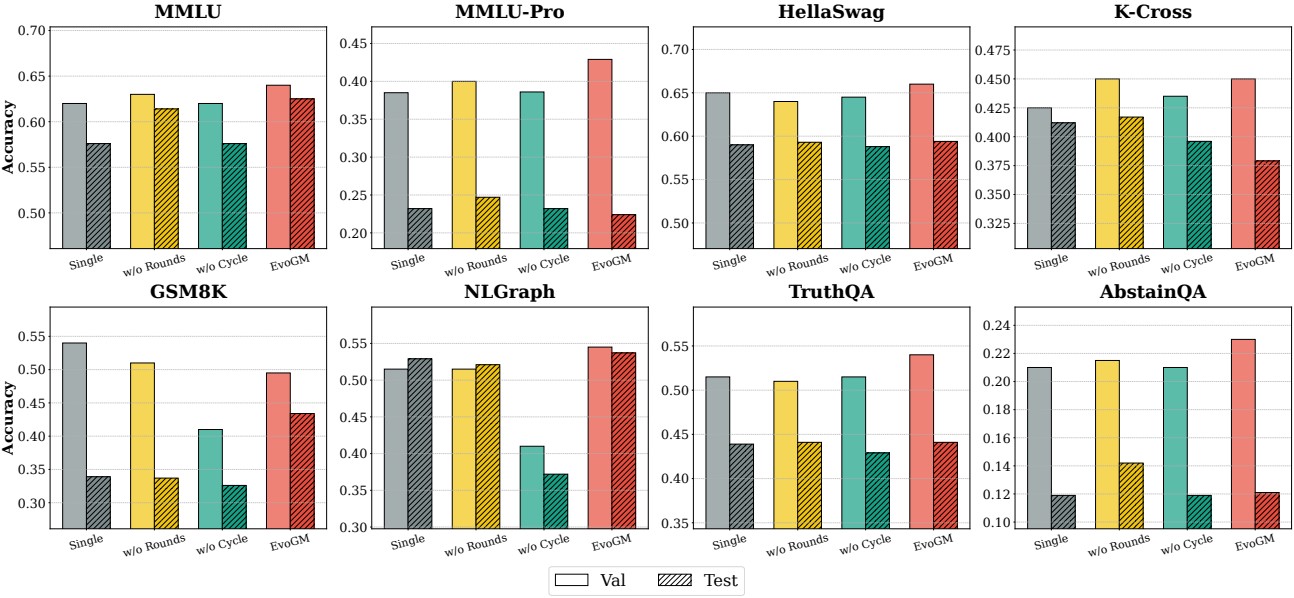

*Figure 7.* PAblation study on different components. (1) Single-Generator represents the single generative model variant; (2) w/o Rounds represents the replacement of multi-round updates with five continuous iterations; and (3) w/o Cycle Loss represents the removal of the cycle-consistency constraint.

*Table 12.* Ablation study on key hyperparameters. The baseline configuration is marked with "–". In the $\Delta$ Test Acc column, the largest improvement is highlighted in **bold**, while performance degradation is shown in *italic*.

| Parameter | Value | Pop | Iter | Epochs | LR | Rounds | Dev Acc | Test Acc | $\Delta$ Test Acc |
|---|---|---|---|---|---|---|---|---|---|
| | 10 | 10 | 5 | 400 | 0.001 | 3 | 0.4954 | 0.4100 | +0.0050 |
| Population Size | 20 | 20 | 5 | 400 | 0.001 | 3 | 0.5104 | 0.4180 | – |
| | 30 | 30 | 5 | 400 | 0.001 | 3 | 0.5129 | 0.4240 | **+0.0305** |
| | 3 | 20 | 3 | 400 | 0.001 | 3 | 0.5200 | 0.4195 | +0.0145 |
| Max Iterations | 5 | 20 | 5 | 400 | 0.001 | 3 | 0.5104 | 0.4180 | – |
| | 8 | 20 | 8 | 400 | 0.001 | 3 | 0.5150 | 0.4245 | +0.0195 |
| | 200 | 20 | 5 | 200 | 0.001 | 3 | 0.5125 | 0.4245 | +0.0115 |
| Generator Epochs | 400 | 20 | 5 | 400 | 0.001 | 3 | 0.5104 | 0.4180 | – |
| | 600 | 20 | 5 | 600 | 0.001 | 3 | 0.5129 | 0.4240 | *-0.0035* |
| | 0.0001 | 20 | 5 | 400 | 0.0001 | 3 | 0.5175 | 0.4270 | +0.0220 |
| Learning Rate | 0.0010 | 20 | 5 | 400 | 0.0010 | 3 | 0.5104 | 0.4180 | – |
| | 0.0050 | 20 | 5 | 400 | 0.0050 | 3 | 0.5175 | 0.4155 | +0.0105 |
| | 1 | 20 | 5 | 400 | 0.001 | 1 | 0.5046 | 0.4015 | *-0.0165* |
| Evolution Rounds | 3 | 20 | 5 | 400 | 0.001 | 3 | 0.5104 | 0.4180 | – |
| | 5 | 20 | 5 | 400 | 0.001 | 5 | 0.5264 | 0.4195 | +0.0130 |

*Table 13.* Performance comparison across different numbers of models and evaluation stages.

| Models | Initial Val | | | | | Models | Final Val | | | | |
|---|---|---|---|---|---|---|---|---|---|---|---|
| | MMLU | MMLU-Pro | HellaSwag | K-Cross | Avg | | MMLU | MMLU-Pro | HellaSwag | K-Cross | Avg |
| 2 | 0.580 | 0.329 | 0.600 | 0.390 | 0.475 | 2 | 0.575 | 0.329 | 0.605 | 0.395 | 0.476 |
| 3 | 0.620 | 0.300 | 0.625 | 0.360 | 0.476 | 3 | 0.615 | 0.314 | 0.640 | 0.365 | 0.484 |
| 4 | 0.585 | 0.314 | 0.635 | 0.375 | 0.477 | 4 | 0.605 | 0.400 | 0.645 | 0.345 | 0.499 |
| 5 | 0.615 | 0.314 | 0.615 | 0.355 | 0.475 | 5 | 0.605 | 0.414 | 0.650 | 0.350 | 0.505 |
| 6 | 0.605 | 0.286 | 0.615 | 0.360 | 0.466 | 6 | 0.615 | 0.400 | 0.660 | 0.345 | 0.505 |
| 7 | 0.605 | 0.286 | 0.595 | 0.370 | 0.464 | 7 | 0.595 | 0.400 | 0.655 | 0.395 | 0.511 |
| 8 | 0.600 | 0.386 | 0.565 | 0.360 | 0.478 | 8 | 0.630 | 0.414 | 0.645 | 0.345 | 0.509 |
| 9 | 0.600 | 0.386 | 0.565 | 0.360 | 0.478 | 9 | 0.615 | 0.386 | 0.655 | 0.375 | 0.508 |
| 10 | 0.600 | 0.386 | 0.565 | 0.360 | 0.478 | 10 | 0.620 | 0.343 | 0.650 | 0.385 | 0.499 |

| Models | Initial Test | | | | | Models | Final Test | | | | |
|---|---|---|---|---|---|---|---|---|---|---|---|
| | MMLU | MMLU-Pro | HellaSwag | K-Cross | Avg | | MMLU | MMLU-Pro | HellaSwag | K-Cross | Avg |
| 2 | 0.559 | 0.196 | 0.578 | 0.400 | 0.433 | 2 | 0.552 | 0.193 | 0.571 | 0.400 | 0.429 |
| 3 | 0.576 | 0.255 | 0.563 | 0.385 | 0.445 | 3 | 0.578 | 0.254 | 0.561 | 0.385 | 0.445 |
| 4 | 0.587 | 0.241 | 0.578 | 0.385 | 0.448 | 4 | 0.601 | 0.250 | 0.590 | 0.386 | 0.457 |
| 5 | 0.579 | 0.251 | 0.585 | 0.391 | 0.452 | 5 | 0.600 | 0.253 | 0.601 | 0.380 | 0.459 |
| 6 | 0.577 | 0.235 | 0.593 | 0.388 | 0.448 | 6 | 0.602 | 0.245 | 0.589 | 0.391 | 0.457 |
| 7 | 0.574 | 0.234 | 0.592 | 0.394 | 0.449 | 7 | 0.589 | 0.234 | 0.599 | 0.393 | 0.454 |
| 8 | 0.582 | 0.232 | 0.575 | 0.344 | 0.433 | 8 | 0.608 | 0.246 | 0.599 | 0.386 | 0.460 |
| 9 | 0.582 | 0.232 | 0.575 | 0.344 | 0.433 | 9 | 0.595 | 0.244 | 0.595 | 0.398 | 0.458 |
| 10 | 0.582 | 0.232 | 0.575 | 0.344 | 0.433 | 10 | 0.582 | 0.228 | 0.590 | 0.405 | 0.451 |

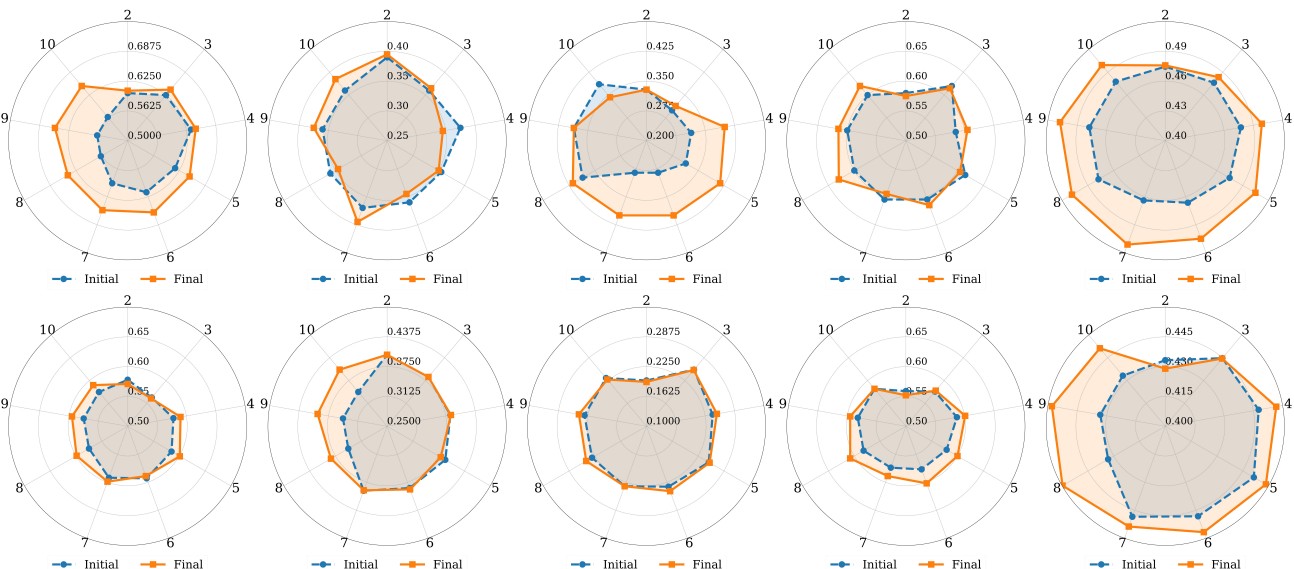

*Figure 8.* Performance Comparison across Multiple Benchmarks. The radar plots illustrate the performance of different model merging methods on Validation (top row) and Test (bottom row) sets. The benchmarks include HellaSwag, Knowledge Crosswords, MMLU Pro, and MMLU, with the rightmost column showing the overall average performance across all tasks.

