# OpenReview forum: "EvoGM: Learning to Merge LLMs via Evolutionary Generative Optimization"
_ICML.cc/2026/Conference — ICML 2026 regular_

### Official Review · Reviewer_84Vn · 2026-03-05

**Soundness:** 3
**Presentation:** 4
**Significance:** 3
**Originality:** 3
**Overall Recommendation:** 5
**Confidence:** 4

**Summary:**

The paper focuses on the task of model merging: given a pre-trained base model and multiple expert models fine-tuned on different tasks, how to obtain a unified model capable of simultaneously handling multiple tasks or generalizing to new tasks by merging the parameters of these models (i.e., task vectors) through weighted averaging. Traditional methods rely on manually designed heuristic rules or random search, which are inefficient and struggle to find the global optimum. The core idea of this paper is to transform the search process for merging coefficients into a learnable generative task.

**Compliance With Llm Reviewing Policy:**

Affirmed.

**Ethics Expertise Needed:**

["Research Integrity Issues (e.g., plagiarism)"]

**Key Questions For Authors:**

1.Your empirical results show that EvoGM outperforms random search methods, but the theoretical explanation for this advantage remains relatively limited. Do you have any insights into why generative search is more effective in the coefficient space? For instance, does the coefficient space exhibit some low-dimensional manifold structure that generative models can capture?
2.Your method assumes all experts are fine-tuned from the same pre-trained base model. Have you considered extending EvoGM to merge models with different architectures or different initializations? If so, what challenges do you foresee?
3.In the multi-task unified model setting, EvoGM finds a single set of coefficients that balances performance across all tasks. However, different tasks may inherently conflict. Have you explored the Pareto frontier of task performance? Could EvoGM be extended to provide users with a range of trade-off options rather than a single solution?

**Limitations:**

The paper focuses on experimental validation, and its theoretical explanation for why generative search outperforms random search in the coefficient space is relatively weak. Questions such as how the cycle-consistency loss helps capture the structure of winning regions and why basis shift can escape local optima remain at the level of empirical observation, lacking deeper theoretical analysis.

**Strengths And Weaknesses:**

The EvoGM paper introduces a novel framework that reformulates model merging as a learnable generative task, combining evolutionary algorithms with dual-generator architectures to search for optimal merging coefficients. Its primary strength lies in methodological innovation—transforming stochastic search into a guided generative process—supported by comprehensive experiments demonstrating state-of-the-art performance across multiple benchmarks, including GLUE tasks and unseen task generalization. The authors provide thorough ablation studies confirming the contribution of each component and show scalability from 2 to 10 expert models.

However, the paper has several limitations. The theoretical foundation remains underdeveloped, offering little explanation for why generative search outperforms random search in coefficient space. Computational cost is significant due to repeated model evaluations, despite parallelization. The method depends heavily on validation set quality and assumes all experts share the same pre-trained base model, restricting applicability to heterogeneous architectures. Multiple hyperparameters require tuning, and the multi-task unified model produces only a single trade-off point without exploring Pareto-optimal solutions. Additionally, comparisons with training-based approaches are limited. Overall, EvoGM represents an important step forward in model merging but leaves room for deeper theoretical analysis and broader practical applicability.

---

> ### Author Rebuttal · Authors · 2026-03-31
>
> Thank you for the positive assessment and for these helpful suggestions. We respond to the main concerns below.
>
> ### Q1. Why does generative search work better than random/stochastic search in the coefficient space?
>
> Our intention is not to claim that the coefficient space follows a clean low-dimensional manifold. A more cautious view is that high-performing regions are not uniformly distributed, and past search trajectories contain useful directional information.
>
> From this perspective, EvoGM improves over random/stochastic search by turning blind perturbation into **history-guided exploration**. Random methods propose new coefficients largely independently, while EvoGM learns from winner/loser trajectories observed during optimization and biases future proposals toward directions that were previously associated with improvement. The optimization-guided objective provides this bias. The cycle-consistency term regularizes the mapping and helps avoid collapsing diverse inputs into nearly identical outputs. The multi-round basis shift serves a different role: when search under a fixed basis starts to saturate, rebuilding the basis from elite merged models changes the local search coordinates and can reopen exploration.
>
> This interpretation is also consistent with our empirical results: EvoGM continues improving after stochastic baselines begin to plateau, and removing cycle consistency or the multi-round design causes clear drops in the ablations.
>
> ### Q2. Can EvoGM be extended to models with different architectures or different initializations?
>
> Thank you for this valuable suggestion. Extending model merging to heterogeneous models is indeed a meaningful direction. However, the main challenge there is not coefficient search itself, but how to construct a reliable merge space across models.
>
> In our formulation, each expert is defined as a task vector relative to a shared pretrained base model, so EvoGM is designed to improve search within this standard merge space. For heterogeneous models, one must first address alignment across different architectures, initializations, or parameter structures before coefficient-based search can be meaningfully defined. Without such alignment, it becomes difficult to tell whether performance differences come from the search method or from the alignment procedure itself.
>
> For this reason, the current paper focuses on the shared-base homogeneous setting, where the merge space is well-defined and the contribution of the search method can be studied cleanly. At the same time, homogeneous merging is not merely a simplified proxy for heterogeneous merging; it is itself a standard and practically important setting in training-free model merging. We therefore view heterogeneous merging as an important future direction, while the present work focuses on improving search quality in the homogeneous setting.
>
>
> ### Q3. Why does the multi-task setting return only one solution instead of a Pareto frontier?
>
> Thank you for this suggestion. An explicit Pareto frontier would indeed be useful when different users prefer different trade-offs across tasks.
>
> In the current paper, however, the multi-task setting is not formulated as a strict multi-objective optimization problem. The goal here is to obtain **one unified and deployable merged model**, so we optimize a single scalar validation objective and return one balanced solution accordingly. This choice matches the practical setting considered in the paper, but it does not aim to characterize the full Pareto frontier.
>
> A related perspective is already visible in the single-task setting: optimizing for different target tasks on the 8-task benchmark leads to different merged models, which indirectly reflects different trade-off preferences. Still, this is different from explicitly modeling and generating Pareto-optimal solutions in the unified setting. Extending EvoGM toward multi-objective or Pareto-style model merging would be a natural and meaningful future direction.
>
> ### Q4. Computational cost, validation data, and training-based comparisons
>
> EvoGM does require repeated evaluation, but in our setting the overhead remains moderate. The main experiments use only a small validation set (200 samples), which is already sufficient to obtain clear gains, and the search process is naturally parallelizable. In our implementation, running all 8 tasks takes about 2.8 hours on 8 NPUs.
>
> For training-based methods, the main focus of this work is training-free model merging, so most baselines are chosen under the same setting. Still, we include the representative training-based baseline MTL. In the multi-task setting, MTL achieves an average score of 0.330, compared with 0.352 for the base model and 0.380 for EvoGM. This suggests that, in our few-shot cross-domain setting, direct joint training can be more affected by inter-task interference, while EvoGM is better matched to the training-free regime studied here.

---

> > ### Author Rebuttal · Reviewer_84Vn · 2026-04-07
> >
> > Thanks for your response. It largely solved my concern. But I still keep my scores.

---

> > > ### Author Response · Authors · 2026-04-07
> > >
> > > Thank you for the feedback and for updating the score. We are glad that the concerns have been addressed. We will incorporate the clarifications into the final version.

---

### Official Review · Reviewer_psJ5 · 2026-03-13

**Soundness:** 3
**Presentation:** 3
**Significance:** 3
**Originality:** 3
**Overall Recommendation:** 4
**Confidence:** 3

**Summary:**

EvoGM proposes replacing hand-crafted stochastic mutation operators in evolutionary model merging with a learned generative model. The core idea is to treat the search for optimal merging coefficients λ ∈ ℝ^N as a generative learning task: two MLPs (G_{-→+} and G_{+→-}) are trained with cycle-consistency and optimization-guided losses on winner-loser pairs extracted from the search history, then used as evolutionary operators. A multi-round "evolving expert basis" mechanism periodically promotes elite merged models as new expert foundations, progressively shifting the search space. Experiments cover FLAN-T5-base (8 experts, GLUE) and Qwen2.5-1.5B (10 experts, 8 unseen tasks) with 8–9 baselines. EvoGM achieves the best average performance on both seen and unseen tasks.

**Compliance With Llm Reviewing Policy:**

Affirmed.

**Final Justification:**

The newly added experiments in the rebuttal has addressed my concerns about performance on large models. I raised my score from 3 to 4

**Key Questions For Authors:**

1. What happens when N is large (e.g., N=30 models)? Does the generative model provide a larger advantage over CMA in higher-dimensional coefficient spaces?
2. What is the total wall-clock time and number of model forward passes for EvoGM vs PSO-Merging for a typical experiment?

**Limitations:**

Yes

**Strengths And Weaknesses:**

## Strengths

- **Novel reformulation.** Casting merging coefficient search as adaptive sampling from a learned distribution is a meaningful conceptual advance over stochastic perturbation-based approaches.
- **Technically sound dual-generator design.** Cycle-consistency prevents mode collapse and ensures the mapping is geometrically consistent; the winner-loser preference signal is a clever way to extract training signal from sparse validation evaluations.
- **Solid ablation.** Testing single-generator, w/o Rounds, and w/o Cycle Loss variants meaningfully isolates each component (Figures 3–4). Each ablation is informative.
- **Broad baseline coverage.** Nine baselines including TA, TIES, DARE, CMA, PSO-Merging, and Model Swarm represent the field well.
- **Multi-setting evaluation.** Seen and unseen task evaluation, two model families, and both single- and multi-task settings provide diverse evidence.

---

## Major Concerns

**[MC1] Only sub-2B models are tested.**
All experiments use FLAN-T5-base (250M) or Qwen2.5-1.5B. Practical model merging is applied to 7B–70B models, where storage and compute costs make training-free merging more valuable. Scaling to larger models may challenge EvoGM's approach: (a) the generator training requires repeated model evaluation, which becomes expensive; (b) the basis shift requires synthesising and storing full merged models each round. The paper does not analyse computational cost relative to baselines or discuss how the method scales.

**[MC2] The search space is too low-dimensional to justify a generative model.**
The merging coefficient space (number of expert model) is ℝ^N where N = 8 or 10 in all experiments. Training two 5-layer MLPs with FC-256 hidden units and cycle-consistency constraints to model distributions over ℝ^8 or ℝ^10 is a substantially over-parameterised approach for this problem. Much simpler methods (covariance matrix adaptation — which the paper already tests as CMA — or even Bayesian optimization with a GP) can model it effectively. The "generative model" framing becomes more compelling when N is large (say, N ≥ 30 models), but it is never tested.

---

> ### Author Rebuttal · Authors · 2026-03-31
>
> Thank you for the constructive review. We respond to the main concerns below.
>
> Note. The original results were obtained on A100 GPUs, while the rebuttal experiments were run on Ascend 910C NPUs. Absolute numbers therefore differ slightly, but the comparison remains fair since all methods were rerun under the same setting.
>
> ### Q1. How does EvoGM perform on larger models?
>
> Thank you for raising this concern. We ran an additional experiment on Qwen3-8B with 10 Tulu-finetuned experts, using the same setup, validation/test split, and merging process as in the paper. The test results are shown below.
>
> | Method | MMLU | MMLU-Pro | HellaSwag | K-Cross | GSM8K | NLGraph | TruthQA | AbstainQA | AVG |
> |:---:|:---:|:---:|:---:|:---:|:---:|:---:|:---:|:---:|:---:|
> | Base | 0.655 | 0.349 | 0.792 | 0.589 | 0.125 | 0.534 | 0.561 | 0.308 | 0.489 |
> | MTL | 0.702 | 0.399 | 0.712 | 0.667 | 0.428 | 0.363 | 0.558 | 0.296 | 0.516 |
> | Single Best | **0.732** | **0.432** | 0.802 | 0.644 | 0.614 | 0.548 | 0.630 | **0.400** | 0.600 |
> | Task Arithmetic | 0.721 | **0.432** | 0.809 | 0.580 | 0.355 | 0.413 | 0.566 | 0.232 | 0.513 |
> | TIES Merging | 0.673 | 0.369 | 0.805 | 0.558 | 0.214 | 0.457 | 0.571 | 0.349 | 0.500 |
> | Model Swarm | 0.712 | 0.420 | 0.789 | **0.669** | 0.588 | 0.551 | **0.637** | 0.381 | 0.593 |
> | **EvoGM (Ours)** | 0.718 | 0.429 | **0.824** | 0.586 | **0.655** | **0.607** | 0.614 | 0.394 | **0.603** |
>
> EvoGM gets the best average result (0.603) on Qwen3-8B, higher than the strongest baseline, Model Swarm (0.593). It also gets the best results on several hard tasks, including HellaSwag (0.824), GSM8K (0.655), and NLGraph (0.607). This shows that EvoGM is not limited to sub-2B models and still works well on a much larger 8B model.
>
> We also measured the practical cost on the same Qwen3-8B setting, using 8 NPUs for all methods. For brevity, we compare with the strongest baseline, Model Swarm. We do not include PSO-Merging here because in our runs under the same setting, it was even slower than Model Swarm.
>
> | Task | EvoGM (Time) | EvoGM (FPs) | Model Swarm (Time) | Model Swarm (FPs) |
> |:---:|:---:|:---:|:---:|:---:|
> | MMLU | 43.1 mins | 160 | 32.5 mins | 100 |
> | MMLU-Pro | 40.4 mins | 160 | 26.2 mins | 100 |
> | HellaSwag | 38.4 mins | 160 | 31.0 mins | 120 |
> | K-Cross | 37.7 mins | 160 | 40.0 mins | 120 |
> | GSM8K | 86.5 mins | 160 | 98.2 mins | 100 |
> | NLGraph | 58.5 mins | 160 | 68.3 mins | 100 |
> | TruthQA | 38.7 mins | 160 | 23.0 mins | 100 |
> | AbstainQA | 42.2 mins | 160 | 109.7 mins | 160 |
> | **Total** | **385.5 mins (~6.4h)** | **1280** | **428.9 mins (~7.1h)** | **900** |
>
> Although EvoGM uses more forward passes overall, its total running time is still competitive, and in this 8B setting it is lower than Model Swarm (6.4h vs. 7.1h). Model Swarm uses fewer forward passes on some tasks because it stops early when the result does not change for several generations.
>
> Scaling to even larger models, such as 370B, is still an important next step. The new 8B results do not fully answer that question, but they already show that EvoGM remains effective and manageable beyond the small-model setting.
>
> ### Q2. What happens when the number of expert models increases?
>
> This is an important question. To test whether the proposed method remains useful as the coefficient space grows, we ran an additional controlled experiment with 20 experts on ViT-B-16, directly comparing CMA-ES and EvoGM under the same 6-iteration budget. We used a fixed subset of 500 validation samples and 1,800 test samples. Both methods were evaluated on exactly the same subset with the same search budget, so the comparison is direct and fair.
>
> | Task | CMA-ES | EvoGM (Ours) |
> |:---:|:---:|:---:|
> | Cars | 0.5028 | **0.7006** |
> | DTD | **0.6528** | 0.5111 |
> | EuroSAT | 0.4828 | **0.8106** |
> | GTSRB | **0.7794** | 0.5061 |
> | MNIST | **0.7050** | 0.5894 |
> | RESISC45 | 0.6767 | **0.7367** |
> | SVHN | **0.8956** | 0.4556 |
> | SUN397 | 0.6206 | **0.6783** |
> | STL10 | 0.9622 | **0.9783** |
> | OxfordIIITPet | **0.9189** | 0.8439 |
> | Flowers102 | 0.6172 | **0.6217** |
> | CIFAR100 | 0.7300 | **0.7611** |
> | PCAM | **0.5744** | 0.5706 |
> | FER2013 | **0.4922** | 0.4394 |
> | CIFAR10 | 0.8511 | **0.9433** |
> | Food101 | 0.6406 | **0.8867** |
> | FashionMNIST | 0.5128 | **0.8044** |
> | RenderedSST2 | 0.4872 | **0.6433** |
> | EMNIST | **0.2939** | 0.1294 |
> | KMNIST | **0.1406** | 0.1161 |
> | **Average** | 0.6268 | **0.6363** |
>
> With 20 experts, EvoGM gets a higher average test result than CMA-ES (0.6363 vs. 0.6268, +0.0095) under the same setting. This suggests that the EvoGM is still useful beyond the low-dimensional cases in the original paper, and can already outperform a strong black-box optimizer in a larger coefficient space.
>
> A more systematic study at even larger scales, such as N = 30+, would still be useful future work. Still, this 20-expert result already shows that EvoGM is not simply over-designed for small N, and that it can continue to help as the search space grows.

---

> > ### Author Rebuttal · Reviewer_psJ5 · 2026-04-02
> >
> > Largely addressed.
> > Will raise the score

---

> > > ### Author Response · Authors · 2026-04-07
> > >
> > > Thank you for the feedback and for updating the score. We are glad that the concerns have been addressed. We will incorporate the clarifications into the final version.

---

### Official Review · Reviewer_s1YL · 2026-03-24

**Soundness:** 3
**Presentation:** 3
**Significance:** 2
**Originality:** 2
**Overall Recommendation:** 4
**Confidence:** 3

**Summary:**

This paper introduces EvoGM, a new way to merge LLMs. Instead of relying on fixed rules or random search, the authors frame the problem as a generative task—essentially learning to find the optimal recipe for combining models. The core of their method is a dual-generator architecture that learns from past successes and failures ("winner-loser pairs"). The most interesting part is an evolutionary loop where the best-merged models from one round become the new "expert" building blocks for the next, which they call a "basis shift." They test this on a range of tasks and show that it outperforms a whole suite of existing merging techniques.

**Compliance With Llm Reviewing Policy:**

Affirmed.

**Final Justification:**

The authors have addressed my main concerns, so I would increase my initial score.

**Key Questions For Authors:**

See the weaknesses 1 and 3.

**Limitations:**

The entire framework, like most current model merging techniques, is built on the assumption that the optimal merged model can be found through a linear combination of task vectors (θ_pre + Σλ_iτ_i). This is a strong constraint. It's entirely possible that the most effective way to combine skills from different experts is non-linear—perhaps requiring more complex, layer-specific, or even neuron-specific combination rules. The paper successfully optimizes for the best λs within this linear space but doesn't question the limitations of the space itself.

**Strengths And Weaknesses:**

## Strengths

1. The author proposes a simple and efficient idea for model merging. Shifting model merging from a simple optimization problem to a learnable, generative one is a genuinely new take. It's a clever way to navigate the tricky parameter landscape and has the potential to open up new research directions.

2. The "Evolving Expert Basis" is a standout feature. By periodically refreshing the pool of "experts" with the best models found so far, the system can co-evolve both its merging strategy and its capabilities. The performance jumps shown in the graphs after each round really sell this point.

## Weaknesses

1. In the "Evolving Expert Basis" section, the author claims it selects the "top-N" coefficients from the history to create the new experts, but it's completely vague on how. With a potentially large history of candidates, how do you pick just N? Is it purely the N best-performing ones? Or do you consider diversity to avoid the new expert pool being too uniform? This is a critical detail, and without it,  nobody could reimplement the method correctly.

2. The paper shows EvoGM outperforming methods like PSO-Merging. However, EvoGM benefits from its multi-round "basis shift" mechanism. Did the baseline methods also get this advantage? For a fair comparison, shouldn't PSO also be allowed to use its best-found models to reset the expert pool at the same interval? Right now, it feels like you're comparing a multi-stage EvoGM against a single-stage version of the baselines, which naturally gives your method an edge.

3. In several tables, the performance gains are tiny (e.g., a 0.5% improvement). While a win is a win, without any confidence intervals or statistical significance tests (like a t-test), it's hard to be sure these small margins aren't just noise from a lucky run. For a paper to claim SOTA, the improvements need to be statistically robust.

4. Some typos, such as: "Specifically,, EvoGM features a dual-generator architecture..." in Page 1, Line 20.
and the formula for the centroid of the winner set is $\mu^+ = \frac{1}{|H^+|} \sum_{\lambda' \in H^+} \lambda'$.  The text uses λ' (lambda prime) to denote the vectors in the winner set H+. However, in the very next formula for L_opt, it uses λ (lambda without the prime) from the loser set H- as input to the generator (G(λ)). While technically correct within each formula's context, constantly switching between λ for losers and λ' for winners can be mentally taxing for the reader.

5. The experiments focus exclusively on merging models from the same family and size (FLAN-T5-base or Qwen2.5-1.5B). A key challenge in model merging is combining models with different architectures or sizes. While the paper doesn't claim to solve this, its silence on the topic implicitly positions EvoGM as another tool for a relatively constrained, homogeneous merging scenario.

---

> ### Author Rebuttal · Authors · 2026-03-31
>
> Thank you for the constructive review. We respond to the main concerns below.
>
> **Note.**  The original results were obtained on A100 GPUs, while the rebuttal experiments were run on Ascend 910C NPUs. Absolute numbers therefore differ slightly, but the comparison remains fair since all methods were rerun under the same setting.
>
> ### Q1. How are the “top-N” coefficients selected to form the new expert basis?
>
> After each round, we select the current top-N candidates by validation performance, and use their merged models as the new experts for the next round. This selection is based on the current population rather than the full search history. We do not explicitly optimize for diversity in this step. We will clarify this in the final version and release the code after publication.
>
> ### Q2. Is the comparison fair, given that EvoGM uses multi-round basis shift?
>
> To check fairness, we ran the suggested control: Multi-Round PSO under the same setup as EvoGM, namely 2 rounds with 3 iterations per round, along with the same basis-reset schedule and overall evaluation budget.
>
> |         Task         | PSO (Single) | PSO (Multi) | EvoGM (Multi) |
> | :------------------: | :----------: | :---------: | :-----------: |
> |         MMLU         |    0.5990    |    0.5660   |     0.5942    |
> |       MMLU-Pro       |    0.2298    |    0.2070   |     0.2412    |
> |       HellaSwag      |    0.5925    |    0.5840   |     0.5930    |
> | Knowledge Crosswords |    0.3853    |    0.3790   |     0.4080    |
> |         GSM8K        |    0.3675    |    0.2060   |     0.4408    |
> |        NLGraph       |    0.5443    |    0.2820   |     0.5262    |
> |      TruthfulQA      |    0.4315    |    0.4279   |     0.4344    |
> |     MMLU-Abstain     |    0.1137    |    0.0880   |     0.1333    |
> |      **Average**    |  **0.4018**  |  **0.3425** |   **0.4214**  |
>
> Simply giving PSO the same multi-round mechanism hurts performance (0.4018 → 0.3425), especially on GSM8K and NLGraph. So basis shift is not a generic trick that automatically benefits any search method.
>
> A likely reason is that vanilla heuristic search quickly concentrates around validation-favored local optima; after basis reset, the expert pool becomes too narrow, hurting later exploration and generalization. EvoGM is less prone to this because it learns from winner/loser trajectories rather than relying only on direct heuristic updates, and cycle consistency further helps avoid overly collapsed mappings. The gain of basis shift in EvoGM therefore comes from its interaction with the generative search mechanism, rather than from an unfair extra advantage.
>
> ### Q3. Are the reported gains statistically significant?
>
> To address robustness, we reran EvoGM with 4 different random seeds and performed **independent two-sample Welch’s t-tests** against PSO and Model Swarm.
>
> |         Task         |  EvoGM (mean ± std) |         PSO         |     Model Swarm     |   vs. PSO (p-value)  |    vs. Swarm (p-value)  |
> | :------------------: | :-----------------: | :-----------------: | :-----------------: | :---------: | :---------: |
> |         MMLU         |   0.5942 ± 0.0184   |   0.5990 ± 0.0072   |   0.6015 ± 0.0163   |    0.6614   |    0.5770   |
> |       MMLU-Pro       |   0.2412 ± 0.0084   |   0.2298 ± 0.0072   |   0.2400 ± 0.0139   |    0.0840   |    0.8839   |
> |       HellaSwag      |   0.5930 ± 0.0074   |   0.5925 ± 0.0115   |   0.5900 ± 0.0262   |    0.9446   |    0.8377   |
> | Knowledge Crosswords |   0.4080 ± 0.0041   |   0.3853 ± 0.0418   |   0.3948 ± 0.0068   |    0.3563   |    0.0211   |
> |         GSM8K        |   0.4408 ± 0.0256   |   0.3675 ± 0.0454   |   0.3295 ± 0.0070   |    0.0400   |    0.0021   |
> |        NLGraph       |   0.5262 ± 0.0184   |   0.5443 ± 0.0506   |   0.3748 ± 0.0005   |    0.5420   |    0.0005   |
> |      TruthfulQA      |   0.4344 ± 0.0159   |   0.4315 ± 0.0184   |   0.4303 ± 0.0148   |    0.8234   |    0.7220   |
> |     MMLU-Abstain     |   0.1333 ± 0.0223   |   0.1137 ± 0.0172   |   0.1330 ± 0.0265   |    0.2190   |    0.9890   |
> |      **Average**     | **0.4214 ± 0.0073** | **0.4018 ± 0.0019** | **0.3867 ± 0.0076** | **0.0100†** | **0.0006‡** |
>
> † p < 0.05, ‡ p < 0.01
>
> Not every single-task difference is significant, but the overall average improvement is significant: EvoGM vs. PSO gives **p = 0.0100†**, and EvoGM vs. Model Swarm gives **p = 0.0006‡**. Significant gains also appear on Knowledge Crosswords (vs. Model Swarm), GSM8K (vs. both PSO and Model Swarm), and NLGraph (vs. Model Swarm). This provides additional evidence that EvoGM’s gains are robust at the overall level.
>
> ### Homogeneous-only setting
>
> See our response to **Reviewer 84Vn, Q2**.
>
> ### Other comments
>
> * The current framework still works within the standard linear task-vector merging space. Exploring nonlinear, layer-wise, or neuron-wise merging is an interesting direction for future work.
> * Thank you for catching the typo and the notation issue around (\lambda) / (\lambda'). We will fix both.

---

> > ### Author Rebuttal · Reviewer_s1YL · 2026-04-02
> >
> > The authors have addressed my main concerns, so I would increase my initial score.

---

> > > ### Author Response · Authors · 2026-04-07
> > >
> > > Thank you for the feedback and for updating the score. We are glad that the concerns have been addressed. We will incorporate the clarifications into the final version.

---

### Decision · Program_Chairs · 2026-04-30

**Decision:**

Accept (regular)

**Comment:**

The paper proposes a novel approach to model merging by formulating the search for merging coefficients as a generative modeling problem. Reviewers generally agree that the method is technically sound, with a well-designed dual-generator architecture and a meaningful reformulation of the search process. The empirical evaluation is thorough, covering multiple benchmarks and baselines, and demonstrates consistent improvements. The rebuttal effectively addressed several key concerns, including fairness of comparisons, statistical significance, and scalability to moderately larger models, leading reviewers to update their scores positively.
The main strengths of the paper lie in its conceptual contribution—casting model merging as a learned generative process—and in its solid empirical validation. At the same time, reviewers noted several limitations, including the restricted experimental setting (homogeneous models with a shared base), limited theoretical understanding of the method, and questions regarding scalability and computational cost.
While the technical contribution is clear, the overall significance is somewhat moderated by the scope of the problem setting and the practical applicability of model merging in real-world pipelines, where alternative approaches (e.g., joint training or routing-based methods) are often used. Nevertheless, within the studied setting, the paper presents a well-executed and novel approach that advances the state of the art.